# EFFECTIVE INTERPLAY BETWEEN SPARSITY AND QUANTIZATION: FROM THEORY TO PRACTICE

**Simla Burcu Harma**
EcoCloud, EPFL
simla.harma@epfl.ch

**Ayan Chakraborty**
EcoCloud, EPFL
ayan.chakraborty@epfl.ch

**Elizaveta Kostenok**
EcoCloud, EPFL
elizaveta.kostenok@epfl.ch

**Danila Mishin**
EcoCloud, EPFL
danila.mishin@epfl.ch

**Dongho Ha**
MangoBoost Inc.
dongho.ha@mangoboost.io

**Babak Falsafi**
EcoCloud, EPFL
babak.falsafi@epfl.ch

**Martin Jaggi**
EcoCloud, EPFL
martin.jaggi@epfl.ch

**Ming Liu**
Google
miliu@google.com

**Yunho Oh**
Korea University
yunho_oh@korea.ac.kr

**Suvinay Subramanian**
Google
suvinay@google.com

**Amir Yazdanbakhsh**
Google DeepMind
ayazdan@google.com

## ABSTRACT

The increasing size of deep neural networks (DNNs) necessitates effective model compression to reduce their computational and memory footprints. Sparsity and quantization are two prominent compression methods that have been shown to reduce DNNs' computational and memory footprints significantly while preserving model accuracy. However, how these two methods interact when combined together remains a key question for developers, as many tacitly assume that they are *orthogonal*, meaning that their combined use does not introduce additional errors beyond those introduced by each method independently. In this paper, we provide the first mathematical proof that sparsity and quantization are non-orthogonal. We corroborate these results with experiments spanning a range of large language models, including the OPT and LLaMA model families (with 125M to 8B parameters), and vision models like ViT and ResNet. We show that the order in which we apply these methods matters because applying quantization before sparsity may disrupt the relative importance of tensor elements, which may inadvertently remove significant elements from a tensor. More importantly, we show that even if applied in the correct order, the compounded errors from sparsity and quantization can significantly harm accuracy. Our findings extend to the efficient deployment of large models in resource-constrained compute platforms to reduce serving cost, offering insights into best practices for applying these compression methods to maximize hardware resource efficiency without compromising accuracy. [1]

## 1 INTRODUCTION

Recent breakthroughs in deep neural networks (DNNs) have surpassed human-level capabilities across various tasks such as text generation, machine translation, and computer vision. Unfortunately, this achievement is accompanied by significant challenges due to the exponential growth in the size and complexity of DNN models and datasets (Brown et al., 2020; Zhang et al., 2022b; Scao et al., 2022; Touvron et al., 2023; Almazrouei et al., 2023; Anil et al., 2023; Jiang et al., 2023; OpenAI, 2023; Mesnard et al., 2024), which complicates their practical deployment and efficient serving. Delivering efficient and real-time inference for these large models is constrained by arithmetic density (throughput/silicon area (Drumond et al., 2018a; Darvish Rouhani et al., 2020a; Harma et al., 2022)), memory footprint, and the pressure on memory bandwidth across various hardware platforms (e.g., GPU (Nvidia, 2022), TPU (Google, 2023)).

---

[1] Code and data are available at: https://sq-interplay.github.io/

Among various efficiency efforts, model compression has emerged as a crucial solution to effectively address the challenges associated with large models (Micikevicius et al., 2018; Drumond et al., 2018b;a; Wang & Kanwar, 2019; Darvish Rouhani et al., 2020b;a; Dai et al., 2021; Zhang et al., 2022a; Yeh et al., 2022; Harma et al., 2022; Rouhani et al., 2023a;b; Hassibi et al., 1993; LeCun et al., 1989; Frantar et al., 2023; Kao et al., 2022; Lasby et al., 2023; Kuzmin et al., 2023) with quantization standing out as a prominent method in terms of overall compression ratio achieved. Quantization effectively reduces the precision of model tensors from native floating-point representation to formats such as FP16 (Micikevicius et al., 2018), BFloat16 (Wang & Kanwar, 2019), and INT8 (van Baalen et al., 2023; Zafrir et al., 2019). In recent years, block-wise numerical formats have gained prominence (Drumond et al., 2018b;a; Darvish Rouhani et al., 2020b;a; Dai et al., 2021; Zhang et al., 2022a; Yeh et al., 2022; Harma et al., 2022; Rouhani et al., 2023a;b), particularly due to their ability to reduce memory footprint and increase arithmetic density in sub-8-bit regimes, while preserving accuracy with minimal hardware overhead.

Beyond quantization, researchers are also exploiting sparsity to further compress models, pruning elements of tensors that are least significant to preserve model accuracy. This reduction in the number of parameters reduces the models' memory footprint and eliminates potentially unnecessary computation (Sze et al., 2020). The most popular sparsity method is magnitude-based sparsity, which prunes elements of a tensor based on their magnitudes (i.e., a proxy for the importance of an element to model accuracy) (Han et al., 2015b; Mishra et al., 2021b; Lu et al., 2023; Ding et al., 2023; Bambhaniya et al., 2024). Magnitude-based sparsity achieves noticeable compression rates with negligible impact on accuracy when combined with fine-tuning (Kurtic et al., 2022; Sanh et al., 2020).

While combining sparsity and quantization provides significant gains in arithmetic density and memory footprint, it may inadvertently have a high impact on model accuracy. Prior work tacitly assumes that these two methods are *orthogonal*, meaning that their combined use does not introduce additional errors beyond those of each method individually. These include studies focusing on CNNs (Han et al., 2015a; Wang et al., 2020), which are more resilient to quantization errors due to the absence of dot-product outliers in activation tensors (Xiao et al., 2023). Because the quantization error in CNNs is relatively low, additional errors introduced by combining the two are minimal. Other studies focus only on compressing weights without quantizing activations (Li et al., 2020; Liu et al., 2023), which also leads to a low overall error when combined with sparsity. These studies fall short of properly investigating the combined impact of these two compression methods to maximize hardware resource efficiency without compromising accuracy.

In this paper, we study the interplay between sparsity and quantization systematically. Sparsity and quantization leverage fundamentally separate computational properties of DNNs, but their combined impact on model accuracy involves complex interactions due to the introduction of errors in tensors. We hypothesize that sparsity and quantization are *non-orthogonal* based on the following two insights. **First**, applying quantization before sparsity (Q→S) may adversely disrupt the relative importance of tensor elements, leading to the removal of significant elements of a tensor with a significant impact on model accuracy. **Second**, applying sparsity before quantization (S→Q) can introduce additional errors in dot product calculations, as these are influenced by the magnitudes and precision of the elements involved, requiring careful investigation. To the best of our knowledge, we are the first to study the interplay between sparsity and quantization in depth to identify the conditions under which accuracy can be preserved or compromised. Our contributions are summarized below:

- **Non-Orthogonality of Sparsity and Quantization**: We prove mathematically that sparsity and quantization are *non-orthogonal* operations. Our per-layer error analysis shows that the combination of the two introduces compounded errors and a degradation of model accuracy. Our findings challenge the conventional wisdom that these methods can be combined without a significant impact on accuracy.

- **Corroborating the Optimal Compression Order**: Although applying sparsity before quantization (S→Q) is the most commonly adopted approach, the optimal order has not been formally demonstrated in the literature. We provide the first mathematical proof that applying sparsity *before* quantization (S→Q) is optimal. Moreover, we derive the upper bound for the error caused by the sub-optimal order of the transformations at the tensor level. We show that it depends linearly on the number of elements in the tensor and the size of the quantization bin.

- **Validating Non-orthogonality Empirically**: We validate our mathematical findings with experiments covering a diverse range of models, including prominent LLMs (OPT, LLaMA), ViT, and ResNet. These experiments support our hypotheses and mathematical findings, underscoring the non-orthogonality of sparsity and quantization, and the optimal order of compression methods. Our experiments demonstrate that combining sparsity and quantization, even in the optimal order, can cause up to 13% additional error in perplexity.

## 2 RELATED WORK

**Quantization.** The ever-growing size of DNN models has spurred extensive research into using narrow numerical formats for inference to reduce memory footprint and improve computational efficiency (Micikevicius et al., 2018; Drumond et al., 2018b;a; Wang & Kanwar, 2019; Darvish Rouhani et al., 2020b;a; Dai et al., 2021; Zhang et al., 2022a; Yeh et al., 2022; Harma et al., 2022; Rouhani et al., 2023a;b). Numerical formats employ scaling factors to adjust their dynamic range and can be categorized based on the granularity and levels of the scaling factors.

Element-wise scaling formats, such as FP32, BFloat16 (Wang & Kanwar, 2019), and FP16 (Micikevicius et al., 2018), consist of sign, mantissa, and exponent components, differing in the bit allocation for each component. Conversely, block-wise scaling formats assign scaling factors to blocks of elements, with block sizes varying by format. For instance, INT8 employs per-tensor scaling, where a single scaling factor is shared by around 1K elements. Recent research highlights the effectiveness of max-scaled fine-grained block-wise scaling formats with block sizes smaller than 100 elements, especially in the sub-8-bit regime for both training and inference (Drumond et al., 2018a; Rouhani et al., 2023a;b; Zhang et al., 2022a; Darvish Rouhani et al., 2020a). Additionally, modern hardware platforms have adopted these techniques. For instance, the upcoming NVIDIA Blackwell GPUs (Nvidia, 2024) will support MXFP.

**Sparsity.** Sparsity methods (Hassibi et al., 1993; LeCun et al., 1989; Frantar et al., 2023; Kao et al., 2022; Lasby et al., 2023) aim to reduce computational and memory footprints in DNNs by selectively pruning tensor elements according to various sparsity mask selection criteria. Broadly, these methods fall into two main categories based on sparsity patterns: unstructured (Han et al., 2015a; Guo et al., 2016; Frankle & Carbin, 2019; Evci et al., 2020a) and structured (Wen et al., 2016; Yao et al., 2019; Kang, 2020; Mishra et al., 2021b; Pool & Yu, 2021; Zhou et al., 2021; Sun et al., 2021; Hubara et al., 2021; Lu et al., 2023). Unstructured sparsity (Han et al., 2015a; Guo et al., 2016; Frankle & Carbin, 2019; Evci et al., 2020a) involves removing individual tensor elements without any specific pattern. Structured sparsity (Wen et al., 2016), on the other hand, employs specific patterns when pruning tensor elements. Recent work (Yao et al., 2019; Kang, 2020) has highlighted the effectiveness of fine-grained N:M structured sparsity in mitigating model accuracy loss.

The fundamental operation in any sparsification scheme is selecting candidate elements for pruning, among which magnitude-based sparsity (Han et al., 2015b) is one of the most widely used methods (Lu et al., 2023; Ding et al., 2023; Bambhaniya et al., 2024). In addition, recent work has introduced one-shot pruning methods, such as SparseGPT (Frantar & Alistarh, 2023) and Wanda (Sun et al., 2023), aiming to eliminate the need for an additional fine-tuning phase. However, evidence suggests that incorporating fine-tuning still improves accuracies significantly (Sun et al., 2023; Syed et al., 2023; Lu et al., 2024).

**Combining sparsity and quantization.** Prior work has studied the combination of sparsity and quantization and its impact on model accuracy in both orders: sparsity followed by quantization (Yu et al., 2023; Frantar & Alistarh, 2023; 2022; Park et al., 2022; Mishra et al., 2021a; Li et al., 2020; Han et al., 2015a) and quantization followed by sparsity (Hu et al., 2021; Hawks et al., 2021; Wu et al., 2023; Mishra et al., 2021b). There are two missing pieces of information from prior work. First, consensus on the optimal order of compression operations is not established. A few studies raise the question and experiment with both orders to determine the best approach (Wu et al., 2023; Wang et al., 2022; Zandonati et al., 2023; Park et al., 2019; Yu et al., 2020; Mishra et al., 2021a; Kozlov et al., 2021; Zhang et al., 2021), while others treat the methods as orthogonal compression schemes. Second, there is a lack of mathematical grasp on how sparsity and quantization errors interact and influence final model performance.

## 3 NON-ORTHOGONALITY OF SPARSITY AND QUANTIZATION

This section provides a mathematical analysis of the interplay between sparsity and quantization, formalizing these compression methods and examining their combination, henceforth referred to as (mathematical) *composition*, at both the tensor and dot-product levels. For the remainder of the paper, we use the following terms: (1) Tensor level refers to structures that encompass both weight and activation tensors; and (2) Dot-product level pertains to the computation of inner products within these tensors, such as the matrix multiplication operation between weights and activations during the forward pass. Our analysis centers on quantization methods that reduce the bit-width of model weights and activations using block-wise numerical formats, which are prevalent in practical implementations (Darvish Rouhani et al., 2020a; Drumond et al., 2018a; Zhang et al., 2022a; Rouhani et al., 2023a;b; Micikevicius et al., 2022). These formats determine the scaling factor based on the element with the maximum magnitude within the block. We refer to any quantization method employing these numerical formats as *max-scaled block-wise quantization*. We use magnitude-based sparsity for both unstructured and N:M structured sparsity.

**Definition 3.1** (**Max-scaled block-wise quantization**). Let $\mathbf{x} \in \mathbb{R}^n$ be a block of $n$ numbers and $m \in \mathbb{N}$ denote the quantization bit-width. Max-scaled block-wise quantization $q : \mathbb{R}^n \to \mathbb{R}^n$ is a transformation of the block $\mathbf{x}$ such that

$$x_i \xrightarrow{q} Q_m(x_i, scale) \tag{1}$$

where $scale = \max(|x_1|, \ldots, |x_n|)$ is the scaling factor. $Q_m(\cdot, scale)$ quantizes the given element with the scaling factor $scale$ and the number of mantissa bits $m$. The exact form of $Q_m$ depends on the numerical format and can be found in Appendix K. For instance, INT$m$ quantization transformation is defined as follows:

$$Q_m(x_i, scale) = s \cdot \left\lfloor \frac{x_i}{s} \right\rceil, \text{where } s = \frac{scale}{2^{m-1} - 1}, \text{ and } \lfloor \cdot \rceil \text{ is the rounding to the nearest integer.} \tag{2}$$

**Definition 3.2** (**Magnitude-based sparsity**). Let $\mathbf{x} \in \mathbb{R}^n$ be a block of $n$ numbers. We assume $n$ is divisible by $M$ and we consider each group of $M$ elements in the block. The magnitude-based N:M sparsity transformation can be formulated as:

$$\tilde{x}_i := \begin{cases} 0 & \text{if } |x_i| < \xi \\ x_i & \text{otherwise} \end{cases}, \text{for } i = 1, 2, ..., M \tag{3}$$

where $\xi$ is the $N$-th largest element in the set $\{|x_1|, \ldots, |x_M|\}$. The same formula can be adjusted to represent $p\%$ unstructured sparsity by defining $\xi$ as the $N$-th largest element in the tensor, where $N = \lfloor M \cdot p/100 \rceil$, $M$ is the number of elements in the tensor, and $\lfloor \cdot \rceil$ is the operation of rounding to the nearest integer.

In the remainder of this section, we delve into the composition of sparsity and quantization at two different levels. First, we examine the effects of applying this composition in different orders at the tensor level, observing how individual tensors are altered. Then, we explore how the composition influences the result of the dot product operation.

### 3.1 TENSOR-LEVEL ANALYSIS

Sparsity and quantization transformations inherently introduce errors by decreasing precision or pruning tensor elements. To study the composition of sparsity and quantization transformations at the tensor level, we introduce formal definitions of *transformation error* and *orthogonality in compression*. We prove that orthogonality in compression between sparsity and quantization *does not* persist within this composition.

The following definition formalizes the error for a specific transformation at the block level, which consists of a subset of tensor elements.

**Definition 3.3** (**Transformation error**). Let $\mathbf{x} \in \mathbb{R}^n$ be a block of $n$ numbers, which are the input of a transformation $f : \mathbb{R}^n \to \mathbb{R}^n$. We define $\varepsilon_f(\mathbf{x}) := \mathbf{x} - f(\mathbf{x})$ as the error of the transformation $f$.

Definition 3.3 can be extended to the tensor level, despite being defined at the block level. The cumulative error of a tensor can be viewed as the summation of individual errors across all its

constituent blocks. Hence, the theorems analyzed at the block level are indicative of the behavior when scaled up to the tensor level.

Composing two compression methods (transformations) is expected to introduce additional errors. Any error introduced by the first transformation becomes part of the input to the second transformation, potentially amplifying the initial error and resulting in a larger overall error.

**Definition 3.4** (**Tensor-level orthogonality**). We define two transformations $f$ and $g$ to be *orthogonal in compression*[2] if any order of their composition does not introduce any additional error, and thus, the following inequalities hold:

$$\forall \mathbf{x} \in \mathbb{R}^n, \ \|\varepsilon_{g \circ f}(\mathbf{x})\| \leq \|\varepsilon_f(\mathbf{x})\| + \|\varepsilon_g(\mathbf{x})\| \ \text{ and } \ \|\varepsilon_{f \circ g}(\mathbf{x})\| \leq \|\varepsilon_f(\mathbf{x})\| + \|\varepsilon_g(\mathbf{x})\| \qquad (4)$$

where $\|\cdot\|$ is an $L_p$ norm, $p \in [1, +\infty)$.

**Theorem 3.5.** Let $q$ be the max-scaled block-wise quantization and $s$ be the magnitude-based sparsity transformation. Applying sparsity before quantization does not introduce any additional error:

$$\forall \mathbf{x} \in \mathbb{R}^n, \ \|\varepsilon_{q \circ s}(\mathbf{x})\| \leq \|\varepsilon_q(\mathbf{x})\| + \|\varepsilon_s(\mathbf{x})\| \qquad (5)$$

Moreover, the equality is attainable.

The proof of Theorem 3.5 can be found in Appendix J. The main idea behind the proof is that as the sparsity transformation does not prune the largest element in the block, the $scale$ quantization parameter remains unchanged. Consequently, the quantization error for the non-zero components before and after sparsity remains the same.

**Theorem 3.6.** Let $q$ be the max-scaled block-wise quantization and $s$ be the magnitude-based sparsity transformation. Applying quantization before sparsity *may* introduce additional error:

$$\exists \mathbf{x} \in \mathbb{R}^n, \ \|\varepsilon_{s \circ q}(\mathbf{x})\| > \|\varepsilon_q(\mathbf{x})\| + \|\varepsilon_s(\mathbf{x})\| \qquad (6)$$

Moreover, a global upper bound exists for the additional error arising from this specific order of transformations (Q→S). This upper bound is solely determined by the quantization method and the parameters of the sparsity type, independent of the input data. The following theorem precisely quantifies the magnitude of this additional error.

**Theorem 3.7.** Let $q$ be the max-scaled block-wise quantization and $s$ be the magnitude-based N:M sparsity transformation. Let $step$ be the least upper bound for the magnitude of the quantization error for one element: $step = \sup\{|\varepsilon_q(\mathbf{x})_i| \mid \mathbf{x} \in \mathbb{R}^n, i \in \{1 \ldots n\}\}$. Then the error of the composition $s \circ q$ with respect to $L_1$ norm has the following upper bound:

$$\forall \mathbf{x} \in \mathbb{R}^n, \ \|\varepsilon_{s \circ q}(\mathbf{x})\|_1 \leq \|\varepsilon_q(\mathbf{x})\|_1 + \|\varepsilon_s(\mathbf{x})\|_1 + \underbrace{2 \cdot step \cdot \frac{M - N}{M} \cdot n}_{\text{additional error}} \qquad (7)$$

The general formulation of Theorem 3.7 for all $L_p$ norms and the proof of Theorem 3.6 and 3.7 can be found in Appendix J. As a corollary of Theorem 3.5, 3.6, and 3.7, it follows that the optimal order of transformations is sparsity followed by quantization, as this sequence *does not* introduce any additional error. Moreover, according to Definition 3.4, sparsity and quantization are *non-orthogonal* at the tensor level.

### 3.2 DOT-PRODUCT-LEVEL ANALYSIS

In this section, we delve into the error linked with the dot product operation, which is the primary operation in DNNs. Our analysis focuses on scenarios where weight tensors undergo sparsity and quantization, while activation tensors solely undergo quantization. We first extend the definition of transformation error to the dot-product level.

**Definition 3.8** (**Transformation error over the dot product**). Let $\mathbf{x}, \mathbf{w} \in \mathbb{R}^n$ denote the inputs of a transformation $f : \mathbb{R}^n \to \mathbb{R}^n$ and the dot product operation $\langle ., . \rangle : \mathbb{R}^n \times \mathbb{R}^n \to \mathbb{R}$. We define $\varepsilon_f^D(\mathbf{x}, \mathbf{w}) := \langle \mathbf{x}, \mathbf{w} \rangle - \langle f(\mathbf{x}), f(\mathbf{w}) \rangle$ as the error of the transformation $f$ over dot product. Similarly, we define $\varepsilon_{f,g}^D(\mathbf{x}, \mathbf{w}) := \langle \mathbf{x}, \mathbf{w} \rangle - \langle f(\mathbf{x}), g(\mathbf{w}) \rangle$ as the error over the dot product when different transformations are applied to $x$ and $w$.

---

[2]In the remainder of the paper, we use the term "orthogonal" to refer to "orthogonal in compression" for simplicity.

At the dot-product level, we define two compression methods as *orthogonal* if their composition, in any order, does not introduce additional error, akin to Definition 3.4.

**Definition 3.9** (**Dot-product-level orthogonality**). Let $\mathbf{x}, \mathbf{w} \in \mathbb{R}^n$ denote the inputs of transformations $f : \mathbb{R}^n \to \mathbb{R}^n$ and $g : \mathbb{R}^n \to \mathbb{R}^n$, and the dot product operation $\langle ., . \rangle : \mathbb{R}^n \times \mathbb{R}^n \to \mathbb{R}$. Let the transformation $f$ be applied to both $\mathbf{x}$ and $\mathbf{w}$, and transformation $g$ be applied only to $\mathbf{w}$. Let $c$ denote a composition of $f$ and $g$ in any order, $c := f \circ g$ or $c := g \circ f$. We define two transformations $f$ and $g$ to be *orthogonal* on the dot-product level if any order of their composition applied to the second term $\mathbf{w}$ does not introduce any additional error:

$$\forall \mathbf{x}, \mathbf{y} \in \mathbb{R}^n, \; |\varepsilon_{f,c}^D(\mathbf{x}, \mathbf{w})| < |\varepsilon_{I,g}^D(\mathbf{x}, \mathbf{w})| + |\varepsilon_f^D(\mathbf{x}, \mathbf{w})| \tag{8}$$

In the following theorem, we demonstrate that any composition of sparsity and quantization yields additional error, rendering these two methods non-orthogonal.

**Theorem 3.10.** Let $q$ be the max-scaled block-wise quantization, $s$ be the magnitude-based sparsity transformation, $c$ be the composition which is either $s \circ q$ or $q \circ s$ and $I$ be the identity function. Composition of max-scaled quantization $q$ and sparsity $s$ in any order produces additional error in any order, given that only the second operand, i.e., weight, is pruned:

$$\exists \mathbf{x}, \mathbf{w} \in \mathbb{R}^n, \; |\varepsilon_{q,c}^D(\mathbf{x}, \mathbf{w})| > |\varepsilon_{I,s}^D(\mathbf{x}, \mathbf{w})| + |\varepsilon_q^D(\mathbf{x}, \mathbf{w})| \tag{9}$$

Moreover,

$$|\varepsilon_{q,c}^D(\mathbf{x}, \mathbf{w})| \leq |\varepsilon_{I,s}^D(\mathbf{x}, \mathbf{w})| + |\varepsilon_q^D(\mathbf{x}, \mathbf{w})| + \underbrace{|\langle q(\mathbf{x}), \tilde{\varepsilon}_c(\mathbf{w}) \rangle|}_{\varepsilon_t} + \underbrace{|\langle \varepsilon_q(\mathbf{x}), \varepsilon_s(\mathbf{w}) \rangle|}_{\varepsilon_i} \tag{10}$$

additional error

where $\tilde{\varepsilon}_c(\mathbf{x})$ is defined as the correction error vector of the composition:

$$\varepsilon_c(\mathbf{x}) = \varepsilon_q(\mathbf{x}) + \varepsilon_s(\mathbf{x}) + \tilde{\varepsilon}_c(\mathbf{x}) \tag{11}$$

Proof of Theorem 3.10 can be found in Appendix J.

**Analysis of the additional error.** As a corollary of Theorem 3.10, the composition of max-scaled sparsity and quantization is *non-orthogonal*, resulting in two additional error terms.

The term $\varepsilon_t$ incorporates the correction vector of the composition $\tilde{\varepsilon}_c$, which carries the additional error from the tensor level to the dot-product level. Depending on the order of the composition, the value of $\varepsilon_t$ varies. When quantization precedes sparsity, certain elements within a block may become equal due to quantization, leading the sparsity step to prune different elements than it would on the original tensor. This introduces additional error, as previously significant elements may be inadvertently pruned. If sparsity precedes quantization, the correction vector $\tilde{\varepsilon}_c$ exclusively comprises the quantization errors of the pruned elements. As a result, the magnitude of the additional error is typically smaller compared to the reverse order.

The term $\varepsilon_i$ also contributes to the additional error, encoding the interaction between the error vectors $\varepsilon_q(\mathbf{x})$ and $\varepsilon_s(\mathbf{w})$. However, as the norm of quantization and sparsity errors are generally smaller than the norm of the quantization block, this term is less significant than $\varepsilon_t$.

We provide a more detailed explanation of the additional error analysis in Appendix M

Finally, to experimentally validate our mathematical findings, we define a metric, *orthogonality threshold* to assess whether the transformations are orthogonal.

**Definition 3.11** (**Orthogonality threshold**). Let $M$ be a DNN model under consideration, $\text{EM}(M)$ be an evaluation metric that measures the performance of the model $M$ (e.g., perplexity or cross-entropy loss), $\text{EM}_C(M)$ be the evaluation metric of the model $M$ with transformation $C$, which is either sparsity $S$ or quantization $Q$. Moreover, let $\text{Err}_C(M) = \text{EM}_C(M) - \text{EM}(M)$ be the evaluation metric error of the transformation $C$ for the model $M$. We define *orthogonality threshold* as:

$$\text{Orthogonality Threshold} = \text{EM}(M) + \text{Err}_Q(M) + \text{Err}_S(M) \tag{12}$$

If the compression methods are non-orthogonal, and the evaluation metric improves with lower values (e.g., perplexity), we expect the compressed model's evaluation metric (e.g., perplexity) to worsen and thus exceed the orthogonality threshold due to compounded errors. Similarly, if the compression methods are non-orthogonal, and the evaluation metric improves with higher values (e.g., accuracy), we expect the compressed model's evaluation metric (e.g., accuracy) to decrease and thus fall below the orthogonality threshold.

# 4 EXPERIMENTAL METHODOLOGY AND RESULTS

**Models, datasets, and evaluation setup.** We study the most widely adopted Transformer-based models, including OPT (Zhang et al., 2022b) and LLaMA (Touvron et al., 2023) model families. In line with prior work (Xiao et al., 2023; Frantar & Alistarh, 2023; Sun et al., 2023), we fine-tune pre-trained models and evaluate perplexity on the WikiText2 (Merity et al., 2017) dataset. The pre-trained LLMs used in our experiments are base (general-purpose) models, not instruct-tuned variants. In addition, we assess non-orthogonality across different metrics of ViT (Dosovitskiy et al., 2021) and ResNet (He et al., 2016) on ImageNet-1k (Deng et al., 2009). In all experiments, we designate the dense FP32 configuration as the primary baseline.

Our experiments span a diverse range of configurations to validate our mathematical findings, including various variants of max-scaled formats, such as INT8 quantization with per channel scaling (Dettmers et al., 2022), HBFP8/6 (Drumond et al., 2018a), and MXFP8/6 (Rouhani et al., 2023b). We primarily study magnitude-based 50% unstructured and 2:4 structured sparsity with sparsity-aware fine-tuning. We define N:M structured sparsity as following: *in every group of M consecutive weights, at most N weights can have non-zero values.* We evaluate the impact of a higher compression ratio on ViT-B/16 and ResNet-50 by applying 75% unstructured sparsity, 1:4 structured sparsity, and HBFP4. Detailed results can be found in Appendix G and O. We also explore post-training one-shot sparsity methods like SparseGPT (Frantar & Alistarh, 2023) and Wanda (Sun et al., 2023) in Appendix F.

In our experiments, we evaluate the impact of each compression method by analyzing the variations in perplexity and cross-entropy loss compared to the baseline, thereby focusing on the cumulative error in the model output. Additionally, we analyze the errors of intermediate layers to support our mathematical analysis, as detailed in Section 4.3 and Appendix I.

Note that we focus on cross-entropy loss for ViT-B/16 and ResNet-50 instead of classification accuracy. This is because accuracy remains unaffected as long as the most likely label remains unchanged, regardless of its absolute value. However, our primary metric for orthogonality threshold is the aggregated errors introduced in the model output distribution. Table 8 in Appendix G and Table 12 in Appendix O present orthogonality threshold on additional metrics.

**Experimental setup.** We exclusively sparsify and/or quantize layers with trainable parameters. Specifically, we target all linear layers in LLMs (excluding the lm-head or embedding layers following the literature (Frantar et al., 2022; Frantar & Alistarh, 2023; Lee et al., 2024)), and all linear and convolution layers (including the initial embedding layer) in ViT-B/16 and ResNet-50. These layers collectively constitute approximately 99% of the total parameters. In all experiments, we prune the weights while keeping the activations dense. Both weights and activation tensors are quantized before matrix multiplication operations. For OPT, LLaMA, ViT, and ResNet fine-tuning, we employ sparse fine-tuning on a dense FP32 pre-trained model, recomputing sparsity masks at each iteration. In experiments involving sparsity followed by quantization (S→Q), we apply one-shot quantization to sparse fine-tuned models. Conversely, for experiments with the reverse order (Q→S), we directly fine-tune the model in a quantized and sparsified manner. At each iteration, we quantize activations and weights while applying sparsity to weight tensors. We validate the effectiveness of these compression recipes through an ablation study, the details of which are presented in Appendix D. To ensure fair comparison, we maintain uniform hyperparameters across various number formats for a given model and sparsity type (details in Appendix B). We present a limited sensitivity study on the initial seed number in Table 9 in Appendix H. A summary of our experimental setup is presented in Table 6.

## 4.1 EMPIRICAL STUDY 1: ORDER OF SPARSITY AND QUANTIZATION

This section presents empirical evidence demonstrating that applying sparsity before quantization leads to better perplexities compared to the reverse order. These results are aligned with the mathematical analysis in Section 3.1. Table 1 presents perplexities for OPT-125M and LLaMA-2-7B under various numerical formats and sparsity types considering both orders of transformations. "*50%*" denotes unstructured sparsity, while "*2:4*" represents a variant of N:M structured sparsity. In the FP32 columns, only one perplexity for each compression order is reported because no quantization is applied. The best results for each (sparsity type, numerical format) pair are highlighted in bold.

Table 1: Model perplexities on WikiText2 for combined sparsity and quantization. The best results for each (sparsity type, number format) pair are highlighted in bold.

| Sparsity type | Order | OPT-125M | | | | | | LLaMA-2-7B | | | | | |
|---|---|---|---|---|---|---|---|---|---|---|---|---|---|
| | | FP32 | INT8 | MXFP8 | MXFP6 | HBFP8 | HBFP6 | FP32 | INT8 | MXFP8 | MXFP6 | HBFP8 | HBFP6 |
| 0% (Dense) | - | 27.65 | 28.06 | 28.45 | 28.01 | 27.81 | 29.91 | 5.12 | 5.15 | 5.17 | 5.16 | 5.12 | 5.24 |
| 50% | S→Q | 29.94 | **30.22** | **31.13** | **31.20** | **30.46** | **32.51** | 6.31 | **6.94** | **6.40** | **6.38** | **6.32** | **6.51** |
| | Q→S | - | 34.71 | 36.39 | 35.60 | 37.48 | 40.86 | - | 8.13 | 8.47 | 9.32 | 9.86 | 10.20 |
| 2:4 | S→Q | 31.89 | **32.76** | **33.99** | **33.41** | **32.25** | **34.58** | 9.30 | **9.37** | **9.35** | **9.32** | **9.39** | **10.68** |
| | Q→S | - | 45.06 | 44.16 | 42.25 | 46.57 | 55.64 | - | 14.65 | 14.35 | 14.50 | 14.98 | 18.64 |

Table 2: Model perplexities and CE loss for combined sparsity and quantization. The numbers in the parentheses show the difference in perplexity/CE loss between the sparse and dense configuration.

| Sparsity type | Number format | OPT-125M | | OPT-6.7B | | LLaMA-2-7B | | LLaMA-3-8B | | ViT-B/16 | |
|---|---|---|---|---|---|---|---|---|---|---|---|
| | | PPL↓ | Orth. threshold | PPL↓ | Orth. threshold | PPL↓ | Orth. threshold | PPL↓ | Orth. threshold | CE Loss↓ | Orth. threshold |
| 0% | FP32 | 27.65 | - | 10.86 | - | 5.12 | - | 5.53 | - | 0.703 | - |
| | INT8 | 28.06 | - | 10.95 | - | 5.19 | - | 5.63 | - | 0.706 | - |
| | MXFP8 | 28.45 | - | 11.25 | - | 5.17 | - | 5.62 | - | 0.722 | - |
| | MXFP6 | 28.01 | - | 11.02 | - | 5.16 | - | 5.62 | - | 0.715 | - |
| | HBFP8 | 27.81 | - | 10.88 | - | 5.12 | - | 5.56 | - | 0.704 | - |
| | HBFP6 | 29.91 | - | 11.20 | - | 5.24 | - | 5.87 | - | 0.718 | - |
| 50% | FP32 | 29.94 (+2.29) | - | 11.30 (+0.44) | - | 6.31 (+1.19) | - | 10.09 (+4.56) | - | 0.723 (+0.020) | - |
| | INT8 | **30.22** (+2.16) | 30.35 | **11.37** (+0.42) | 11.39 | 6.94 (+1.75) | **6.38** | 10.85 (+5.22) | **10.19** | 0.728 (+0.022) | **0.725** |
| | MXFP8 | 31.13 (+2.68) | **30.74** | 11.74 (+0.49) | **11.69** | 6.40 (+1.23) | **6.36** | 10.34 (+4.72) | **10.18** | 0.745 (+0.023) | **0.742** |
| | MXFP6 | 31.20 (+3.19) | **30.30** | 11.53 (+0.51) | **11.44** | 6.38 (+1.22) | **6.35** | **10.15** (+4.53) | 10.18 | **0.734** (+0.019) | 0.735 |
| | HBFP8 | 30.46 (+2.65) | **30.18** | 11.31 (+0.43) | **11.32** | 6.32 (+1.20) | **6.31** | 10.12 (+4.56) | **10.12** | 0.724 (+0.020) | **0.723** |
| | HBFP6 | 32.51 (+2.60) | **32.2** | 11.94 (+0.74) | **11.65** | 6.51 (+1.27) | **6.43** | 10.55 (+4.68) | **10.43** | **0.736** (+0.018) | 0.737 |
| 2:4 | FP32 | 31.89 (+4.24) | - | 15.48 (+4.62) | - | 9.30 (+4.18) | - | 13.07 (+7.54) | - | 0.759 (+0.056) | - |
| | INT8 | 32.76 (+4.70) | **32.30** | 15.61 (+4.66) | **15.57** | **9.37** (+4.18) | **9.37** | 13.23 (+7.60) | **13.17** | 0.762 (+0.056) | **0.761** |
| | MXFP8 | 33.99 (+5.54) | **32.69** | **15.70** (+4.45) | 15.87 | **9.35** (+4.18) | **9.35** | 13.35 (+7.73) | **13.16** | 0.781 (+0.059) | **0.777** |
| | MXFP6 | 33.41 (+5.40) | **32.25** | 15.95 (+4.93) | **15.64** | **9.32** (+4.16) | 9.34 | 13.20 (+7.58) | **13.16** | **0.770** (+0.055) | 0.771 |
| | HBFP8 | 32.25 (+4.44) | **32.05** | 15.57 (+4.69) | **15.50** | 9.39 (+4.27) | **9.31** | 13.11 (+7.55) | **13.1** | 0.760 (+0.056) | **0.759** |
| | HBFP6 | 34.58 (+4.67) | **34.15** | 16.98 (+5.78) | **15.82** | 10.68 (+5.44) | **9.42** | 13.64 (+7.77) | **13.41** | 0.774 (+0.056) | **0.773** |

We evaluate the order of transformations across different configurations: sparsity followed by quantization (S→Q) and quantization followed by sparsity (Q→S). Configurations with lower perplexities are highlighted in bold. Consistently, we observe that the S→Q order yields better perplexities across all numerical formats for magnitude-based sparsity. As discussed in Section 3.1, in the case of Q→S, quantizing a tensor can alter the order of its elements due to changes in magnitudes. If magnitude-based sparsity prunes elements of the quantized tensor that were originally larger before quantization, the error of the combination may exceed the sum of the errors caused by each transformation individually. This compounded error further propagates through subsequent dot-product and vector operations, impacting overall model performance.

## 4.2 Empirical study 2: Non-Orthogonality between sparsity and quantization

This section demonstrates that combining sparsity and quantization results in additional error, surpassing the sum of their individual errors. Table 2 presents perplexities for OPT-125M, OPT-6.7B, LLaMA-2-7B and LLaMA-3-8B on WikiText2, and cross-entropy (CE) loss for ViT-B/16 on ImageNet-1k for various combinations of numerical formats and sparsity types. Following our conclusions regarding the order of transformations, we only report results for S→Q. We compute the orthogonality threshold for each combination by summing the individual errors from sparsity and quantization relative to the baseline dense FP32 model, using Equation 12. Each model's performance is compared against this bound, with superior results highlighted in bold. In the majority of configurations, perplexity and cross-entropy loss values exceed the orthogonality thresholds, validating the non-orthogonality of sparsity and quantization.

We mathematically show (Section 3.2) that the additional error introduced by combining sparsity and quantization significantly depends on the values of quantized activation tensors and the quantization error of sparsified weights. This error tends to amplify through successive dot products and vector operations. Consequently, the quantization error affects the additional error caused by the combination

more than the sparsity error. The gap between the model's performance and the orthogonality threshold is minimal for numerical formats with minimal performance decrease, whereas larger errors are observed for formats with larger errors. For instance, HBFP6 results in a `2.26` increase in perplexity for OPT-125M, while its combination with 50% unstructured sparsity leads to a `4.86` increase.

Our analysis reveals that both model size and compression ratio significantly influence the additional error introduced by combining sparsity and quantization. Larger models exhibit greater tolerance to these compression methods, resulting in lower additional errors. Moreover, formats with minimal quantization errors (e.g. MXFP8), and sparsity types with minimal sparsification errors (e.g. unstructured sparsity) lead to lower additional errors, even for smaller models. The effect of high quantization error is more pronounced for sparsity types known for higher errors. For instance, HBFP6's combination with 2:4 sparsity causes a `6.93` perplexity increase for OPT-125M and a `6.12` perplexity increase for OPT-6.7B. In contrast, combining HBFP6 with unstructured sparsity results in smaller increases of `4.86` and `0.79` for the respective models.

We also observe instances where the orthogonality threshold is slightly higher than the actual perplexity: (a) INT8 with 50% unstructured sparsity for both OPT models, (b) MXFP8 with 2:4 sparsity for OPT-6.7B, (c) MXFP6 with 2:4 sparsity for LLaMA-2-7B, and (d) MXFP6 50% unstructured sparsity for LLaMa-3-8B. Although these occurrences do not consistently correlate with specific formats, sparsity types, or model sizes, they do not contradict our mathematical analysis, which primarily concerns upper bounds of errors and does not entirely rule out orthogonal cases. Furthermore, orthogonal configurations still result in larger errors compared to applying either sparsity or quantization alone. Our mathematical analysis (Theorem 3.10) indicates that there *exists at least one occurrence* where the orthogonality is *not* preserved. This underscores the need for careful examination when applying these compression methods together, as they do not guarantee high accuracies. Delving into cases where orthogonality is preserved falls beyond the scope of this paper, and we leave it for future work.

We observe that although the cross-entropy loss for ViT-B/16 is higher than the calculated orthogonality threshold in most cases in Table 2, the difference is relatively small. We hypothesize the reason behind this behavior is due to the fact that ViT-B/16, being a vision model that operates on images, is more robust to sparsity and quantization errors than LLMs that operate on text. Hence, the sparsity and quantization levels shown in Table 2 are not sufficient to induce large errors, and hide the non-orthogonality of sparsity and quantization. To test our hypothesis, we increase the compression further by employing 75% unstructured sparsity, 1:4 structured sparsity and HBFP4 on ViT-B/16 (Appendix G). The results show that the difference between the baseline cross-entropy loss and the calculated orthogonality threshold for these cases increases significantly, validating the non-orthogonality of sparsity and quantization. This increase in cross-entropy loss also translates to a significant non-orthogonal drop in accuracy. Additionally, we run the same experiments on a CNN-based vision model, ResNet-50 (Appendix O), and the experimental results corroborate these findings.

## 4.3 ABLATION: ERROR PROPAGATION ACROSS LAYERS

In this section, we study error propagation across the layers of a deep neural network.

While our mathematical study provides insights into how sparsity and quantization introduce errors at the dot-product and tensor levels, mathematically estimating the cumulative effect of these errors on the model's final loss or performance is non-trivial. Therefore, we perform a layer-wise empirical analysis to verify that the order of applying sparsity and quantization not only affects per-layer errors but also significantly impacts overall model's accuracy. By inspecting the intermediate outputs of the pre-trained OPT-125M model, we show that the error accumulation increases with the layer index and that the choice of the order directly influences the final model performance.

We apply quantization and magnitude-based sparsity to all linear layers of the pre-trained OPT-125M model in a zero-shot manner. We feed a sample from the test subset through both the compressed model and the corresponding full precision dense model, and we measure the difference in Feed-Forward outputs for each Transformer block. We repeat this experiment for both the S→Q and Q→S orders, and we compare the $L_2$ errors at each layer. The results are shown in Figure 1.

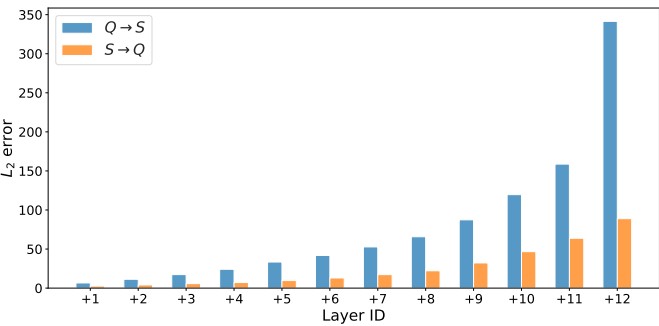

Figure 1: Cumulative error for the combination of 2:4 structured sparsity and HBFP6 quantization

As illustrated in the figure, the per-layer error increases consistently with the layer index, regardless of the order of transformations, and reaches its peak in the final layer. However, we observe that the S→Q order produces significantly lower errors at each intermediate layer compared to the reverse order. This pattern indicates that the error introduced by Q→S accumulates more rapidly as layers deepen, highlighting the detrimental effects of applying quantization before sparsity.

## 5 DISCUSSION

Our mathematical analysis and experimental results offer multiple insights for ML model practitioners. First, our analysis demonstrates a risk-free method to improve model performance, measured by lower perplexity and/or higher accuracy, through choosing the optimal ordering of compression operations for any max-scaled number format and magnitude-based pruning scheme. This contribution is particularly important in the current ML landscape, where sparsity and quantization are pivotal methods for reducing the memory footprint and bandwidth requirements of state-of-the-art LLMs. Second, we show that calculating the orthogonality threshold offers a close enough estimate of model performance (e.g. accuracy, perplexity, etc.) under conditions of sparsity and quantization. This bound can streamline the search for optimal sparse-quantized model configurations by effectively narrowing the search space. There is an inherent trade-off between the hardware benefits of various sparse-quantized configurations and the achieved model performance. Quantization bit-width and sparsity level are key factors influencing the memory and bandwidth requirements for serving these models. For example, at a 50% sparsity level, 8-bit and 6-bit quantization result in total reductions in memory footprint and bandwidth requirements by $8\times$ and $10.7\times$, respectively.

Ideally, practitioners aim to maximize compression (increase sparsity ratio and/or reduce the average bit-width per element). Our analysis elucidates the individual and combined impacts of these factors across a range of recent large models, providing practical guidelines to achieve the highest compression without compromising model performance. Typically, 8-bit quantization with any max-scaled number format can serve as a direct replacement for FP32 when combined with any form of sparsity in the optimal order (S→Q). As discussed in Section 4, certain models exhibit sensitivity to sub-8bit number formats and structured sparsity combinations, even when applied in an optimal order. In scenarios where improvements in arithmetic density (TOPS/mm$^2$) and memory footprint justify a slight reduction in model performance, such as the deployment of large models on edge devices, these combinations may still be viable.

In this work, we do not consider heterogeneous sparsity and quantization schemes, where the sparsity fraction and quantization bit-width vary across layers and differ between activation and weight tensors. Such approaches (Rouhani et al., 2023b; Ma et al., 2024; Harma et al., 2022; Evci et al., 2020b) have demonstrated to be effective in maintaining model accuracy or perplexity while improving compressing ratio. However, these schemes may not be hardware-friendly, introduce noticeable overhead, and are impractical to implement on off-the-shelf hardware platforms (e.g. GPU, TPU). We leave the investigation of the interactions between these heterogeneous sparsity and quantization schemes for future work.

## 6 CONCLUSION

We provide a comprehensive analysis of the interplay between sparsity and quantization in DNNs, showing that applying sparsity before quantization (S→Q) minimizes additional errors and yields better model accuracy. Moreover, our mathematical analysis and extensive empirical study with large language models (OPT, LLaMA), vision transformers (ViT), and convolutional neural networks (ResNet) demonstrate that sparsity and quantization are *non-orthogonal* and their combined use can adversely affect model accuracy. Our findings provide valuable insights for optimizing the compression of large models while preserving accuracy.

### ACKNOWLEDGMENTS

We extend our gratitude towards Zhifeng Chen, Cliff Young, James Laudon, and Shashwat Shrivastava for reviewing the paper and providing insightful feedback. We also thank the extended team at Google DeepMind, who enabled and supported this research direction. This work was partially supported by the SNSF project "Unified Accelerators for Post-Moore Machine Learning" (200021_212757) and a Microsoft Research PhD Fellowship. A part of this paper was the result of a research project supported by SK hynix Inc.

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

# Appendix

## Table of Contents

## A ADDITIONAL RELATED WORK

**Quantization.** Element-wise scaling formats, such as FP32, BFLOAT32 (Wang & Kanwar, 2019), and FP16 (Micikevicius et al., 2018), consist of sign, mantissa, and exponent components, differing in the bit allocation for each component. Conversely, block-wise scaling formats assign scaling factors to blocks of elements, with block sizes varying by format. For instance, INT8 employs per-tensor scaling, where a single scaling factor is shared by around 1K elements.

Recent research highlights the effectiveness of fine-grained block-wise scaling formats with block sizes smaller than 100 elements, especially in the sub-8-bit regime for both training and inference (Drumond et al., 2018a; Rouhani et al., 2023a;b; Zhang et al., 2022a; Darvish Rouhani et al., 2020a). These formats are further categorized into single-level and two-level scaling. Single-level block-wise formats, such as HBFP (Drumond

Table 3: Types of max-scaled numerical formats

| | | Element-wise | | Single-level block-wise | | | Two-level block-wise | |
|---|---|---|---|---|---|---|---|---|
| | | FP32/FP16 | BFloat16 | INT | HBFP | MXINT | MXFP | FP8 |
| Scaling level 1 | Block size | 1 | 1 | 1k | 64 | 32 | 32 | 10k |
| | Scale type | HW | HW | SW | HW | HW | HW | SW |
| Scaling level 2 | Block size | - | - | - | - | - | 1 | 1 |
| | Scale type | - | - | - | - | - | HW | HW |

et al., 2018a; Darvish Rouhani et al., 2020a) and MXINT (Rouhani et al., 2023b), enable fixed-point arithmetic by sharing a single exponent within a block of mantissa or integers. Two-level formats, like MXFP (Rouhani et al., 2023a;b; Micikevicius et al., 2022) and FP8, use more granular scaling factors at the second level, offering greater robustness across diverse range of models.

**Sparsity.** Unstructured sparsity (Han et al., 2015a; Guo et al., 2016; Frankle & Carbin, 2019; Evci et al., 2020a) involves removing individual tensor elements without any specific pattern. Structured sparsity (Wen et al., 2016), on the other hand, employs specific patterns when pruning tensor elements. Recent work (Yao et al., 2019; Kang, 2020) has highlighted the effectiveness of fine-grained N:M structured sparsity in mitigating model accuracy loss. The introduction of the 2:4 structured-sparse Tensor Core in the Nvidia Ampere architecture (Nvidia, 2021) has further driven research in developing N:M sparsity training recipes (Mishra et al., 2021b; Pool & Yu, 2021; Zhou et al., 2021; Sun et al., 2021; Hubara et al., 2021; Lu et al., 2023).

The fundamental operation in any sparsification scheme is selecting candidate elements for pruning among which magnitude-based sparsity (Han et al., 2015b) is one of the most widely used methods (Lu et al., 2023; Ding et al., 2023; Bambhaniya et al., 2024). In addition, recent work has introduced one-shot pruning methods, such as SparseGPT (Frantar & Alistarh, 2023) and Wanda (Sun et al., 2023), aiming to eliminate the need for an additional fine-tuning phase. While these methods achieve state-of-the-art performance for one-shot pruning, evidence suggests that incorporating a fine-tuning phase can lead to better model quality (Sun et al., 2023; Syed et al., 2023). Although these methods are proposed to eliminate fine-tuning and present state-of-the-art accuracies for one-shot pruning, it has been shown that fine-tuning still improves accuracies significantly (Sun et al., 2023; Syed et al., 2023).

# B HYPERPARAMETERS

We perform full parameter fine-tuning while applying magnitude-based sparsity methods. We find the optimal hyperparameters through grid search for each model and sparsity type and apply the same hyperparameters across all number formats, including FP32. We observe that fine-tuning in a Q→S order, where we quantize and sparsify tensors at each iteration, leads to a highly unstable training process, especially with the structured sparsity. For this reason, we impose limitations on the number of training iterations and the learning rate. Thus, we prioritize achieving reproducible and comparable results across all number formats over achieving full convergence for each specific configuration.

Table 4: Details of the sparse fine-tuning experiments

| Model | Sparsity type | Batch size | Weight decay | Optimizer | FT num. iterations | Learning rate |
|---|---|---|---|---|---|---|
| OPT-125M | 50% | 8 | - | Adam | 1776 | $1e^{-4}$ |
| | 2:4 | 8 | - | Adam | 1776 | $1e^{-4}$ |
| OPT-6.7B | 50% | 4 | - | Adam | 1000 | $5e^{-4}$ |
| | 2:4 | 4 | - | Adam | 1500 | $5e^{-4}$ |
| LLaMA-2-7B | 50% | 2 | $1e^{-3}$ | AdamW | 150 | $2e^{-4}$ |
| | 2:4 | 2 | $1e^{-3}$ | AdamW | 60 | $5e^{-5}$ |
| LLaMA-3-8B | 50% | 2 | $1e^{-3}$ | AdamW | 200 | $2e^{-5}$ |
| | 2:4 | 1 | $1e^{-3}$ | AdamW | 300 | $2e^{-5}$ |
| ViT-B/16 | 50% | 32 | $1e^{-3}$ | Adam | 30000 | $5e^{-5}$ |
| | 2:4 | 32 | $1e^{-3}$ | Adam | 30000 | $5e^{-5}$ |

OPT-125M and OPT-6.7B models are fine-tuned with block sizes of 512 and 1024 respectively, while LLaMa models utilize a block size of 2048. All configurations employ a linear learning rate schedule without a warm-up.

## C  COMPUTE RESOURCES AND RUNTIME

We conduct our experiments on four NVIDIA A100 GPUs with 80GB memory, and for small models, we use four NVIDIA V100 GPUs with 32GB memory. The hyperparameters used in our experiments, including the number of fine-tuning epochs, are given in Appendix B. In summary, the estimated runtime for each fine-tuning experiments on these hardware platforms are as follows: (a) 20 minutes for OPT-125M, (b) 5-6 hours for OPT-6.7B, (c) 2-3 hours for LLaMA-2-7B and LLaMA-3-8B, and (d) 40 hours for ViT-B/16.

## D  FINE-TUNING STRATEGIES

Magnitude-based sparsity applied in one-shot causes a significant perplexity degradation and thus needs an additional fine-tuning to recover the perplexity. In combination with quantization, several fine-tuning strategies are possible:

1. Sparse fine-tuning of FP32 model followed by the post-training quantization, sparsity masks are applied to the FP32 weight tensors

2. Fine-tuning in sparsified and quantized manner, where we sparsify and then quantize tensors at each iteration

3. Sparse fine-tuning of FP32 model followed by the post-training quantization, sparsity masks are applied to the quantized weight tensors

4. Fine-tuning in quantized and sparsified manner, where we quantize and then sparsify tensors at each iteration

The former two strategies correspond to the S→Q order of transformations, while the latter two correspond to the Q→S order. We conduct ablation experiments for the OPT-125M to compare these fine-tuning strategies. The results are presented in the Table 5.  According to our results, post-training quantization outperforms

Table 5: Validation perplexities of OPT-125M on WikiText2, produced by different fine-tuning strategies. The best results for each configuration are highlighted in bold.

| Sparsity type | Number format | Order | Quantization during fine-tuning | PPL |
|---|---|---|---|---|
| 50% | HBFP8 | S→Q | × | **30.46** |
| | | | ✓ | 33.51 |
| | | Q→S | × | 39.04 |
| | | | ✓ | **37.48** |
| 50% | HBFP6 | S→Q | × | **32.51** |
| | | | ✓ | 36.20 |
| | | Q→S | × | 41.97 |
| | | | ✓ | **40.86** |

sparse-and-quantized fune-tuning in S→Q order. In contrast, fine-tuning in quantized and sparsified manner recovers perplexity better than post-training quantization in the reverse order.

## E  EXPERIMENTAL SETUP SUMMARY

Table 6 presents a summary of our experimental setup.

## F  POST-TRAINING ONE-SHOT SPARSITY METHODS

Other sparsity schemes, such as Wanda and SparseGPT, have a different pruning policy, which uses activations to assess the significance of the weights and prunes only the least significant ones. The pruning metrics in Wanda and SparseGPT are

$$S_{ij} = |\mathbf{W}_{ij}| \cdot \|\mathbf{X}_j\|_2 \quad \text{and} \quad S_{ij} = [|\mathbf{W}|^2 / \text{diag}((\mathbf{X}^T\mathbf{X} + \lambda\mathbf{I})^{-1}]_{ij} \tag{13}$$

respectively. If quantization is applied before sparsity, the input values will change, which might also change the set of the nullified weights. However, the significance of those weights will not change considerably. Therefore, the correction vector $\boldsymbol{t}_c$ consists of the least significant weights, which are multiplied by the elements of $q(\mathbf{x})$

**Compressed operations by model type**

| | LLMs | ViTs | CNNs |
|---|---|---|---|
| Linear Layers | Yes | Yes | Yes |
| Convolution Layers | – | – | Yes |
| Embedding Layer | No | Yes | – |
| Attention MatMuls | Yes | Yes | – |

**Tensor-level granularity for compressed operations**

| | Weights | Activations | Gradients |
|---|---|---|---|
| Forward Pass (for inference and finetuning) | Sparsified + quantized | Quantized | – |
| Backward Pass (for finetuning only) | Quantized | Quantized | Quantized |
| (1) Note: Only matrix multiplications are compressed; other operations (e.g., optimizer updates) are in FP32 | | | |
| (2) Note: Master weights are stored in FP32, since our primary aim is to compress inference. | | | |

**Workflow for the two different compression orders**

| | |
|---|---|
| S → Q | - Sparse finetuning followed by zero-shot quantization
- Sparsity mask recomputed for each iteration |
| Q → S | - Quantization and sparse finetuning
- Quantization and sparsity masks recomputed for each iteration |

Table 6: Summary of experimental setup

with the lowest values due to the chosen pruning metrics. As a result, the magnitude of $\varepsilon_t$ and the effect of changing the order of the operations is much lower for those sparsity schemes than for the magnitude-based sparsity.

Table 7: Model perplexities on WikiText2 for combined sparsity and quantization. The numbers in the parentheses show the difference in perplexity between the sparse and dense configuration.

| Sparsity type | Sparsity method | Order | OPT-125M (↓) | | | | | | LLaMA-2-7B (↓) | | | | | |
|---|---|---|---|---|---|---|---|---|---|---|---|---|---|---|
| | | | FP32 | INT8 | MXFP8 | MXFP6 | HBFP8 | HBFP6 | FP32 | INT8 | MXFP8 | MXFP6 | HBFP8 | HBFP6 |
| 0% | - | - | 27.65 | 28.06 | 28.45 | 28.01 | 27.81 | 29.91 | 5.12 | 5.15 | 5.17 | 5.16 | 5.12 | 5.24 |
| 50% | Magnitude | S→Q | $29.94^{(+2.29)}$ | $30.22^{(+2.16)}$ | $31.13^{(+2.68)}$ | $31.20^{(+3.19)}$ | $30.46^{(+2.65)}$ | $32.51^{(+2.60)}$ | $6.31^{(+1.19)}$ | $6.94^{(+1.79)}$ | $6.40^{(+1.23)}$ | $6.38^{(+1.22)}$ | $6.32^{(+1.2)}$ | $6.51^{(+1.27)}$ |
| | | Q→S | - | $34.71^{(+6.65)}$ | $36.39^{(+7.94)}$ | $35.60^{(+7.59)}$ | $37.48^{(+9.67)}$ | $40.86^{(+10.95)}$ | - | $8.13^{(+2.98)}$ | $8.47^{(+3.30)}$ | $9.32^{(+4.16)}$ | $9.86^{(+4.74)}$ | $10.20^{(+4.96)}$ |
| | Wanda | S→Q | $38.97^{(+11.32)}$ | $39.29^{(+11.23)}$ | $39.72^{(+11.27)}$ | $40.02^{(+12.01)}$ | $39.21^{(+11.40)}$ | $42.33^{(+12.42)}$ | $6.46^{(+1.34)}$ | $6.47^{(+1.32)}$ | $6.53^{(+1.36)}$ | $6.53^{(+1.37)}$ | $6.48^{(+1.36)}$ | $6.73^{(+1.49)}$ |
| | | Q→S | - | $40.01^{(+11.95)}$ | $40.58^{(+12.13)}$ | $40.43^{(+12.42)}$ | $40.52^{(+12.71)}$ | $42.57^{(+12.66)}$ | - | $6.46^{(+1.31)}$ | $6.55^{(+1.38)}$ | $6.52^{(+1.36)}$ | $6.48^{(+1.36)}$ | $6.79^{(+1.55)}$ |
| | SparseGPT | S→Q | $33.24^{(+5.59)}$ | $33.22^{(+5.16)}$ | $35.27^{(+6.82)}$ | $34.22^{(+6.21)}$ | $33.41^{(+5.60)}$ | $35.86^{(+5.95)}$ | $6.51^{(+1.39)}$ | $6.51^{(+1.36)}$ | $6.58^{(+1.41)}$ | $6.58^{(+1.42)}$ | $6.52^{(+1.40)}$ | $6.77^{(+1.53)}$ |
| | | Q→S | - | $33.54^{(+5.48)}$ | $35.32^{(+6.87)}$ | $34.29^{(+6.28)}$ | $33.64^{(+5.83)}$ | $36.80^{(+6.89)}$ | - | $6.53^{(+1.38)}$ | $6.60^{(+1.43)}$ | $6.58^{(+1.42)}$ | $6.55^{(+1.43)}$ | $6.93^{(+1.69)}$ |
| 2:4 | Magnitude | S→Q | $31.89^{(+4.24)}$ | $32.76^{(+4.7)}$ | $33.99^{(+5.54)}$ | $33.41^{(+5.40)}$ | $32.25^{(+4.44)}$ | $34.58^{(+4.67)}$ | $9.30^{(+4.18)}$ | $9.37^{(+4.22)}$ | $9.35^{(+4.18)}$ | $9.32^{(+4.16)}$ | $9.39^{(+4.27)}$ | $10.68^{(+5.44)}$ |
| | | Q→S | - | $45.06^{(+17.00)}$ | $44.16^{(+15.71)}$ | $42.25^{(+14.24)}$ | $46.57^{(+18.76)}$ | $55.64^{(+25.73)}$ | - | $14.65^{(+9.50)}$ | $14.35^{(+9.18)}$ | $14.50^{(+9.34)}$ | $14.98^{(+9.86)}$ | $18.64^{(+13.40)}$ |
| | Wanda | S→Q | $79.91^{(+52.26)}$ | $79.81^{(+51.75)}$ | $85.25^{(+56.80)}$ | $84.10^{(+56.09)}$ | $80.62^{(+52.81)}$ | $90.66^{(+60.75)}$ | $11.36^{(+6.24)}$ | $11.37^{(+6.22)}$ | $11.15^{(+5.98)}$ | $11.35^{(+6.19)}$ | $11.45^{(+6.33)}$ | $12.74^{(+7.50)}$ |
| | | Q→S | - | $80.28^{(+52.22)}$ | $86.69^{(+58.24)}$ | $84.38^{(+56.37)}$ | $80.69^{(+52.88)}$ | $91.04^{(+61.13)}$ | - | $11.28^{(+6.13)}$ | $11.24^{(+6.07)}$ | $11.46^{(+6.30)}$ | $11.36^{(+6.24)}$ | $13.61^{(+8.37)}$ |
| | SparseGPT | S→Q | $45.14^{(+17.49)}$ | $45.34^{(+17.28)}$ | $48.44^{(+19.99)}$ | $46.49^{(+18.48)}$ | $45.52^{(+17.71)}$ | $50.74^{(+20.83)}$ | $10.22^{(+5.10)}$ | $10.21^{(+5.06)}$ | $10.15^{(+4.98)}$ | $10.26^{(+5.10)}$ | $10.26^{(+5.14)}$ | $10.86^{(+5.62)}$ |
| | | Q→S | - | $44.96^{(+16.9)}$ | $48.67^{(+20.22)}$ | $46.50^{(+18.49)}$ | $45.82^{(+18.01)}$ | $57.39^{(+27.48)}$ | - | $10.21^{(+5.06)}$ | $10.24^{(+5.07)}$ | $10.26^{(+5.10)}$ | $10.21^{(+5.09)}$ | $11.16^{(+5.92)}$ |

We further explore the effectiveness of post-training one-shot sparsity methods, specifically SparseGPT and Wanda, which utilize a selection criterion based on the product of the magnitudes of weights and activations. We report our results in Table 7. We observe that magnitude-based sparsity continues to achieve better perplexities due to fine-tuning. However, because of their selection criterion, SparseGPT and Wanda are not affected by the order of the operations. Even when quantization alters the relative magnitudes within a weight tensor, the corresponding activations can compensate by preserving the original ranking of importance when multiplied together. Consequently, the difference in perplexities between S→Q and Q→S for these methods is minimal, and in few instances, Q→S yields better perplexities.

## G  ORTHOGONALITY THRESHOLD FOR ViT

Table 8 shows the results of cross-entropy loss across various combinations of sparsity and quantization schemes for ViT-B/16 performing the image classification task on ImageNet-1k. First, we note that the calculated orthogonality threshold serves as a correct lower bound for most configurations, supporting our mathematical analysis. Second, ViT-B/16 is significantly more robust to the combination of sparsity and quantization schemes compared to the other LLMs studied in this paper. When used with moderate sparsity levels (50% and 2:4) and 8-bit/6-bit number formats, the actual cross-entropy loss is close to the calculated orthogonality threshold, showing the robustness of ViT-B/16. Only at higher compression rates achieved by using 75% or 1:4 sparsity and 4-bit number formats such as HBFP4, do we see the impact of the sparsity and quantization errors affecting the final cross-entropy loss, making it significantly higher than the calculated orthogonality threshold.

Table 8: Comparison of evaluation cross-entropy loss with estimated orthogonality thresholds.

| Sparsity type | Number format | ViT-B/16 | | | |
| | | Metric | | Orthogonality Threshold | |
| | | Accuracy | CE Loss | Accuracy | CE Loss |
|---|---|---|---|---|---|
| 0% | FP32 | 81.70% | 0.703 | - | - |
| | INT8 | 81.64% | 0.706 | - | - |
| | MXFP8 | 81.12% | 0.722 | - | - |
| | MXFP6 | 81.26% | 0.715 | - | - |
| | HBFP8 | 81.67% | 0.704 | - | - |
| | HBFP6 | 81.35% | 0.718 | - | - |
| | HBFP4 | 72.73% | 1.094 | - | - |
| 50% | FP32 | 81.04% | 0.723 | - | - |
| | INT8 | **81.03%** | 0.728 | 80.98% | **0.725** |
| | MXFP8 | **80.50%** | 0.745 | 80.46% | **0.742** |
| | MXFP6 | **80.80%** | **0.734** | 80.60% | 0.735 |
| | HBFP8 | 81.00% | 0.724 | **81.01%** | **0.723** |
| | HBFP6 | 80.64% | **0.736** | **80.69%** | 0.737 |
| | HBFP4 | **73.38%** | **1.058** | 72.07% | 1.113 |
| 2:4 | FP32 | 80.06% | 0.759 | - | - |
| | INT8 | 79.95% | 0.762 | **80.00%** | **0.761** |
| | MXFP8 | 79.48% | 0.781 | **79.48%** | **0.777** |
| | MXFP6 | **79.73%** | **0.770** | 79.62% | 0.771 |
| | HBFP8 | **80.06%** | 0.760 | 80.03% | **0.759** |
| | HBFP6 | 79.69% | 0.774 | **79.71%** | **0.773** |
| | HBFP4 | 71.06% | 1.163 | **71.09%** | **1.149** |
| 75% | FP32 | 77.26% | 0.881 | - | - |
| | INT8 | 77.03% | 0.897 | **77.20%** | **0.884** |
| | MXFP8 | 76.57% | 0.913 | **76.68%** | **0.900** |
| | MXFP6 | **76.99%** | 0.895 | 76.82% | **0.894** |
| | HBFP8 | 77.14% | **0.882** | **77.23%** | **0.882** |
| | HBFP6 | 76.89% | 0.899 | **76.91%** | **0.896** |
| | HBFP4 | 66.84% | 1.365 | **68.29%** | **1.272** |
| 1:4 | FP32 | 73.24% | 1.055 | - | - |
| | INT8 | 72.90% | 1.070 | **73.18%** | **1.058** |
| | MXFP8 | 72.36% | 1.095 | **72.66%** | **1.074** |
| | MXFP6 | 72.92% | 1.070 | **72.80%** | **1.068** |
| | HBFP8 | 73.16% | 1.057 | **73.21%** | **1.056** |
| | HBFP6 | 72.77% | 1.078 | **72.89%** | **1.070** |
| | HBFP4 | 59.58% | 1.725 | **64.27%** | **1.446** |

# H ROBUSTNESS ANALYSIS

To substantiate our conclusions on the optimal compression operation order, we conduct limited experiments across three distinct random seeds. We report the mean perplexities and error bars in the Table 9. Given the computational cost of fine-tuning, we limit the robustness analysis to the OPT-125M model and HBFP8/6 number format. For both sparsity types we observe stable results, consistently affirming the higher efficacy of the S→Q order. Note that deviations causes by different seeds do *not* compromise the integrity of our conclusions.

Table 9: Validation perplexities of OPT-125M on WikiText2 for S→Q and Q→S. We report mean and standard deviation over three random seeds.

| Sparsity Type | Number Format | Order | PPL |
|---|---|---|---|
| 50% | HBFP8 | S→Q | 30.5 (± 0.2) |
| | | Q→S | 37.4 (± 0.3) |
| | HBFP6 | S→Q | 32.5 (± 0.2) |
| | | Q→S | 40.8 (± 0.3) |
| 2:4 | HBFP8 | S→Q | 32.2 (± 0.1) |
| | | Q→S | 46.5 (± 0.4) |
| | HBFP6 | S→Q | 34.6 (± 0.2) |
| | | Q→S | 55.5 (± 0.8) |

# I LAYER-WISE ERROR PROPAGATION

We also examine the error propagation to ensure that errors introduced in earlier layers do not vanish in subsequent layers. Figure 2 illustrates how compression errors persist and propagate throughout the pre-trained OPT-2.7b model. In this analysis, we quantize each layer in isolation and measure the corresponding propagated error to the other layers, which remain in full precision. We observe that relatively large errors introduced in earlier layers, despite small fluctuations, stay on the same level as they go through the network. As a result, even a single-layer error can significantly degrade model performance, highlighting the potential threats of errors due to the non-orthogonality of compression techniques or applying them in the sub-optimal order.

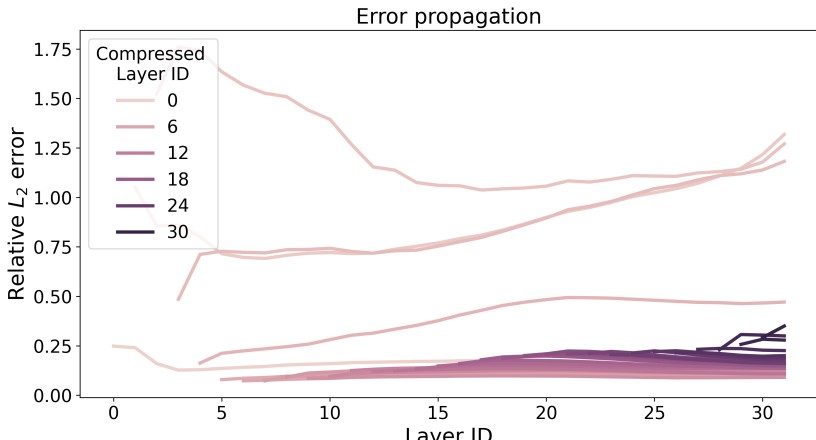

Figure 2: Error dynamics for single-layer quantization. Each line represents the relative $L_2$ error of outputs ($\|\hat{Y_i} - Y_i\|_2 / \|Y_i\|_2$) for the compressed layer at the particular index and all subsequent layers, which remain in full precision.

## J PROOFS FOR THE MATHEMATICAL ANALYSIS

*Proof of Theorem 3.5.* Let $n_s$ represent the number of elements pruned from the block by the sparsity transformation. Without loss of generality, we assume that the last $n_s$ elements in the block are pruned, as permuting the elements does not affect the block's norm. As the sparsity transformation does not prune the largest element in the block, the *scale* parameter of quantization remains unchanged. Consequently, the quantization error for the non-zero components before and after sparsity remains the same:

$$
\|\varepsilon_{q \circ s}(\mathbf{x})\| = \left\| \begin{pmatrix} x_1 \\ \vdots \\ x_{n-n_s} \\ x_{n-n_s+1} \\ \vdots \\ x_n \end{pmatrix} - q \begin{pmatrix} x_1 \\ \vdots \\ x_{n-n_s} \\ 0 \\ \vdots \\ 0 \end{pmatrix} \right\| = \left\| \begin{pmatrix} \varepsilon_q(\mathbf{x})_1 \\ \vdots \\ \varepsilon_q(\mathbf{x})_{n-n_s} \\ x_{n-n_s+1} \\ \vdots \\ x_n \end{pmatrix} \right\| \leq \tag{14}
$$

$$
\leq \left\| \begin{pmatrix} \varepsilon_q(\mathbf{x})_1 \\ \vdots \\ \varepsilon_q(\mathbf{x})_{n-n_s} \\ 0 \\ \vdots \\ 0 \end{pmatrix} \right\| \left\| \begin{pmatrix} 0 \\ \vdots \\ 0 \\ x_{n-n_s+1} \\ \vdots \\ x_n \end{pmatrix} \right\| = \left\| \begin{pmatrix} \varepsilon_q(\mathbf{x})_1 \\ \vdots \\ \varepsilon_q(\mathbf{x})_{n-n_s} \\ 0 \\ \vdots \\ 0 \end{pmatrix} \right\| + \left\| \begin{pmatrix} \varepsilon_s(\mathbf{x})_1 \\ \vdots \\ \varepsilon_s(\mathbf{x})_{n-n_s} \\ \varepsilon_s(\mathbf{x})_{n-n_s+1} \\ \vdots \\ \varepsilon_s(\mathbf{x})_n \end{pmatrix} \right\| \leq \tag{15}
$$

$$
\leq \left\| \begin{pmatrix} \varepsilon_q(\mathbf{x})_1 \\ \vdots \\ \varepsilon_q(\mathbf{x})_{n-n_s} \\ \varepsilon_q(\mathbf{x})_{n-n_s+1} \\ \vdots \\ \varepsilon_q(\mathbf{x})_n \end{pmatrix} \right\| + \left\| \begin{pmatrix} \varepsilon_s(\mathbf{x})_1 \\ \vdots \\ \varepsilon_s(\mathbf{x})_{n-n_s} \\ \varepsilon_s(\mathbf{x})_{n-n_s+1} \\ \vdots \\ \varepsilon_s(\mathbf{x})_n \end{pmatrix} \right\| = \|\varepsilon_s(\mathbf{x})\| + \|\varepsilon_q(\mathbf{x})\| \tag{16}
$$

For $L_p$ norms, where $p \in (1, +\infty)$, the upper bound is attainable under one of two conditions: either the pruned elements in $\mathbf{x}$ are originally zero, or the quantization error for all elements of $\mathbf{x}$ is zero.

In the first case, the first inequality becomes an equality because $\forall i \in \{n - n_s + 1, \ldots, n\} : x_i = 0$. In the second case, the first inequality also becomes an equality because $\forall i \in \{1, \ldots, n - n_s\} : \varepsilon_q(\mathbf{x})_i = 0$.

Similarly, the second inequality becomes an equality as quantization maps zero to zero. Thus, the quantization error for elements in $\{n - n_s + 1, \ldots, n\}$ is zero either because the quantization error for all elements of $\mathbf{x}$ is zero or because the elements were originally zero: $\forall i \in \{n - n_s + 1, \ldots, n\} : \varepsilon_q(\mathbf{x})_i = 0$.

For $L_1$ norm there exist a non-trivial case. We consider the block of floating-point numbers $\mathbf{x} = (4.0, 4.1)^T$, INT4 quantization and $1:2$ sparsity.

$$
s(\mathbf{x}) = \begin{pmatrix} 0.0 \\ 4.1 \end{pmatrix} \quad q(\mathbf{x}) = \begin{pmatrix} 4.0 \\ 4.0 \end{pmatrix} \quad q(s(\mathbf{x})) = \begin{pmatrix} 0.0 \\ 4.0 \end{pmatrix} \tag{17}
$$

The L1-norms of the transformation errors are the following:

$$
\|\varepsilon_s(\mathbf{x})\|_1 = \left\| \begin{pmatrix} 4.0 \\ 0.0 \end{pmatrix} \right\|_1 = 4.0 \quad \|\varepsilon_q(\mathbf{x})\|_1 = \left\| \begin{pmatrix} 0.0 \\ 0.1 \end{pmatrix} \right\|_1 = 0.1 \quad \|\varepsilon_{q \circ s}(\mathbf{x})\|_1 = \left\| \begin{pmatrix} 4.0 \\ 0.1 \end{pmatrix} \right\|_1 = 4.1 \tag{18}
$$

Therefore, $\|\varepsilon_{q \circ s}(\mathbf{x})\|_1 = \|\varepsilon_q(\mathbf{x})\|_1 + \|\varepsilon_s(\mathbf{x})\|_1$ is attainable. $\square$

*Proof of Theorem 3.6.* Consider the block of floating-point numbers $\mathbf{x} = (3.9, 4.0)^T$, INT4 quantization $q$ and $1:2$ sparsity $s$. After applying the quantization transformation to the block, initial relation between its elements $x_i < x_j$ is no longer preserved and both elements have equal probability to be zeroed out by the sparsity transformation. If sparsity zeroes out the element that was initially larger, the resulting error can exceed the sum of the errors caused by each transformation individually:

$$
s(\mathbf{x}) = \begin{pmatrix} 0.0 \\ 4.0 \end{pmatrix} \quad q(\mathbf{x}) = \begin{pmatrix} 4.0 \\ 4.0 \end{pmatrix} \quad s(q(\mathbf{x})) = \begin{pmatrix} 4.0 \\ 0.0 \end{pmatrix} \tag{19}
$$

$$
\|\varepsilon_s(\mathbf{x})\| = \left\| \begin{pmatrix} 3.9 \\ 0.0 \end{pmatrix} \right\| \quad \|\varepsilon_q(\mathbf{x})\| = \left\| \begin{pmatrix} -0.1 \\ 0.0 \end{pmatrix} \right\| \quad \|\varepsilon_{s \circ q}(\mathbf{x})\| = \left\| \begin{pmatrix} -0.1 \\ 4.0 \end{pmatrix} \right\| \tag{20}
$$

We consider $L_p$ norms, where $p \in [1; +\infty)$. In these norms, $\forall a \in \mathbb{R} : \|(a, 0)^T\| = |a| \cdot \|(1, 0)^T\| = |a|$. Therefore:

$$\|\varepsilon_q(\mathbf{x})\| + \|\varepsilon_s(\mathbf{x})\| = \left\| \begin{pmatrix} -0.1 \\ 0.0 \end{pmatrix} \right\| + \left\| \begin{pmatrix} 3.9 \\ 0.0 \end{pmatrix} \right\| = |-0.1| + |3.9| = 4.0 = \tag{21}$$

$$= \left\| \begin{pmatrix} 0.0 \\ 4.0 \end{pmatrix} \right\| < \left\| \begin{pmatrix} -0.1 \\ 4.0 \end{pmatrix} \right\| = \|\varepsilon_{s \circ q}(\mathbf{x})\| \tag{22}$$

Thus, for this particular input $\mathbf{x}$, the inequality $\|\varepsilon_{s \circ q}(\mathbf{x})\| > \|\varepsilon_q(\mathbf{x})\| + \|\varepsilon_s(\mathbf{x})\|$ holds true. $\qquad \square$

**Theorem J.1** (Upper-bound of the error for sub-optimal order, general case). Let $q$ be the max-scaled block-wise quantization and $s$ be the magnitude-based N:M sparsity transformation. Let $step$ be the least upper bound for the magnitude of the quantization error for one element: $step = \sup\{|\varepsilon_q(\mathbf{x})_i| \mid \mathbf{x} \in \mathbb{R}^n, i \in \{1 \ldots n\}\}$. Let $\vec{\mathbf{1}}(n, N, M) \in \mathbb{R}^n$ be a vector with $\frac{M-N}{M} \cdot n$ ones and $\frac{N}{M} \cdot n$ zeros in any order. Then the error of the composition $s \circ q$ with respect to $L_1$ norm has the following upper bound:

$$\forall \mathbf{x} \in \mathbb{R}^n, \ \|\varepsilon_{s \circ q}(\mathbf{x})\|_1 \le \|\varepsilon_q(\mathbf{x})\| + \|\varepsilon_s(\mathbf{x})\| + \underbrace{2 \cdot step \cdot \|\vec{\mathbf{1}}(n, N, M)\|}_{\text{additional error}} \tag{23}$$

*Proof of Theorem J.1.* Without loss of generality for simplicity we assume that the sparsity operation nullifies the last elements within the vector.

$$\|\varepsilon_{s \circ q}(\mathbf{x})\| = \left\| \begin{pmatrix} x_1 \\ \vdots \\ x_{n-n_s} \\ x_{n-n_s+1} \\ \vdots \\ x_n \end{pmatrix} - s \begin{pmatrix} x_1 - \boldsymbol{q}(\mathbf{x})_1 \\ \vdots \\ x_{n-n_s} - \boldsymbol{q}(\mathbf{x})_{n-n_s} \\ x_{n-n_s+1} - \boldsymbol{q}(\mathbf{x})_{n-n_s+1} \\ \vdots \\ x_n - \boldsymbol{q}(\mathbf{x})_n \end{pmatrix} \right\| = \left\| \begin{pmatrix} \varepsilon_q(\mathbf{x})_1 \\ \vdots \\ \varepsilon_q(\mathbf{x})_{n-n_s} \\ x_{n-n_s+1} \\ \vdots \\ x_n \end{pmatrix} \right\| \le \tag{24}$$

$$\le \left\| \begin{pmatrix} \varepsilon_q(\mathbf{x})_1 \\ \vdots \\ \varepsilon_q(\mathbf{x})_{n-n_s} \\ 0 \\ \vdots \\ 0 \end{pmatrix} \right\| + \left\| \begin{pmatrix} 0 \\ \vdots \\ 0 \\ x_{n-n_s+1} \\ \vdots \\ x_n \end{pmatrix} \right\| \le \left\| \begin{pmatrix} \varepsilon_q^1 \\ \vdots \\ \varepsilon_q^{n-n_s} \\ \varepsilon_q^{n-n_s+1} \\ \vdots \\ \varepsilon_q^n \end{pmatrix} \right\| + \left\| \begin{pmatrix} 0 \\ \vdots \\ 0 \\ x_{n-n_s+1} \\ \vdots \\ x_n \end{pmatrix} \right\| = \tag{25}$$

$$= \|\varepsilon_q(\mathbf{x})\| + \left\| \begin{pmatrix} 0 \\ \vdots \\ 0 \\ x_{n-n_s+1} \\ \vdots \\ x_n \end{pmatrix} \right\| := \|\boldsymbol{q}(\mathbf{x})\| + \|\widetilde{\boldsymbol{s}}(\mathbf{x})\| \tag{26}$$

When quantization is applied first, two distinct numbers can become the same: $x_i < x_j \to \boldsymbol{q}(\mathbf{x})_i = \boldsymbol{q}(\mathbf{x})_j$. When we sparsify the quantized numbers, the number that was smaller might get nullified, as depicted in Figure 3. Therefore, the last component $\widetilde{\boldsymbol{s}}$ of the upper bound does not equal $\boldsymbol{s}$.

However, in this case we can get an upper bound for the distance between them:

$$\begin{cases} x_i < x_j \\ \boldsymbol{q}(\mathbf{x})_i \ge \boldsymbol{q}(\mathbf{x})_j \end{cases} \Leftrightarrow \begin{cases} x_i < x_j \\ x_i - \boldsymbol{q}(\mathbf{x})_i \ge x_j - \boldsymbol{q}(\mathbf{x})_j \end{cases} \Leftrightarrow \begin{cases} x_j - x_i > 0 \\ x_j - x_i \le \boldsymbol{q}(\mathbf{x})_j - \boldsymbol{q}(\mathbf{x})_i \end{cases} \Rightarrow \tag{27}$$

$$\Rightarrow |x_j - x_i| \le |\boldsymbol{q}(\mathbf{x})_j - \boldsymbol{q}(\mathbf{x})_i| \le |\boldsymbol{q}(\mathbf{x})_j| + |\boldsymbol{q}(\mathbf{x})_i| \le 2 \cdot step \tag{28}$$

For each $x_i$ that was nullified after quantization followed by sparsification we define $x_i^t$ to be an element that would be nullified, if quantization was not applied. Then the vector $\boldsymbol{s}(\mathbf{x})$ consists of all and only such elements $x_i^t$.

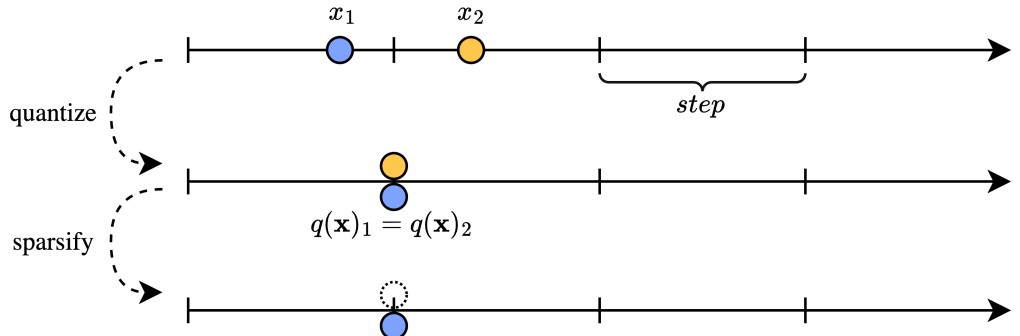

Figure 3: A visual representation of applying quantization first and then sparsification. After quantization two distinct elements become equal. Then, when sparsification is applied, the element that was originally bigger gets nullified as the sparsification operation cannot differentiate them by their magnitude.

There exists a permutation $W$ of the vector $\boldsymbol{s}(\mathbf{x})$ such that maps the element $x_i$ in $\widetilde{\boldsymbol{s}}(\mathbf{x})$ to the element $x_i^t$ in $\boldsymbol{s}(\mathbf{x})$. Therefore, an upper bound for $\|\widetilde{\boldsymbol{s}}(\boldsymbol{x})\|$ is:

$$\|\widetilde{\boldsymbol{s}}(\mathbf{x})\| = \|\widetilde{\boldsymbol{s}}(\mathbf{x}) - W\boldsymbol{s}(\mathbf{x}) + W\boldsymbol{s}(\mathbf{x})\| \leq \|W\boldsymbol{s}(\mathbf{x})\| + \|\widetilde{\boldsymbol{s}}(\mathbf{x}) - W\boldsymbol{s}(\mathbf{x})\| = \tag{29}$$

$$= \|\boldsymbol{s}(\mathbf{x})\| + \left\| \begin{pmatrix} 0 - 0 \\ \vdots \\ 0 - 0 \\ x_{n-n_s+1} - x_{n-n_s+1}^t \\ \vdots \\ x_n - x_n^t \end{pmatrix} \right\| \leq \|\boldsymbol{s}(\mathbf{x})\| + \left\| \begin{pmatrix} 0 \\ \vdots \\ 0 \\ 2 \cdot step \\ \vdots \\ 2 \cdot step \end{pmatrix} \right\| \tag{30}$$

For the case of $N : M$ sparsity the number of nullified elements within the block equals $\frac{M-N}{M} \cdot n$. Therefore:

$$\left\| \begin{pmatrix} 0 \\ \vdots \\ 0 \\ 2 \cdot step \\ \vdots \\ 2 \cdot step \end{pmatrix} \right\| = 2 \cdot step \cdot \left\| \begin{pmatrix} 0 \\ \vdots \\ 0 \\ 1 \\ \vdots \\ 1 \end{pmatrix} \right\| = 2 \cdot step \cdot \|\vec{\mathbf{1}}(n, N, M)\| \tag{31}$$

As a result, the upper bound for the error of the composition is the following:

$$\|\varepsilon_{s \circ q}(\mathbf{x})\| \leq \|\boldsymbol{q}(\mathbf{x})\| + \|\widetilde{\boldsymbol{s}}(\mathbf{x})\| \leq \|\varepsilon_q(\mathbf{x})\| + \|\varepsilon_s(\mathbf{x})\| + 2 \cdot step \cdot \|\vec{\mathbf{1}}(n, N, M)\| \tag{32}$$

$\square$

*Proof of Theorem 3.7.* As a corollary of Theorem J.1, with respect to $L_1$ norm, the last error term can be evaluated as follows:

$$2 \cdot step \cdot \|\vec{\mathbf{1}}(n, N, M)\|_1 = 2 \cdot step \cdot \left\| \begin{pmatrix} 0 \\ \vdots \\ 0 \\ 1 \\ \vdots \\ 1 \end{pmatrix} \right\|_1 = 2 \cdot step \cdot \frac{M-N}{M} \cdot n \tag{33}$$

$\square$

*Proof of Theorem 3.10.* Error of the composition can be written as the following:

$$\varepsilon_{q,c}^{D}(\mathbf{x}, \mathbf{w}) = \langle \mathbf{x}, \mathbf{w} \rangle - \langle q(\mathbf{x}), c(\mathbf{w}) \rangle \tag{34}$$

$$= \langle \mathbf{x}, \mathbf{w} \rangle - \langle \mathbf{x} - \varepsilon_q(\mathbf{x}), \mathbf{w} - \varepsilon_c(\mathbf{w}) \rangle \tag{35}$$

$$= \langle \varepsilon_q(\mathbf{x}), \mathbf{w} \rangle + \langle \mathbf{x}, \varepsilon_c(\mathbf{w}) \rangle - \langle \varepsilon_q(\mathbf{x}), \varepsilon_c(\mathbf{w}) \rangle \tag{36}$$

$$= \langle \varepsilon_q(\mathbf{x}), \mathbf{w} \rangle + \langle \mathbf{x}, \varepsilon_q(\mathbf{w}) + \varepsilon_s(\mathbf{w}) + \tilde{\varepsilon}_c(\mathbf{w}) \rangle - \langle \varepsilon_q(\mathbf{x}), \varepsilon_q(\mathbf{w}) + \varepsilon_s(\mathbf{w}) + \tilde{\varepsilon}_c(\mathbf{w}) \rangle \tag{37}$$

$$= \underbrace{\langle \mathbf{x}, \varepsilon_s(\mathbf{w}) \rangle}_{\varepsilon_{I,s}^{D}(\mathbf{x}, \mathbf{w})} + \underbrace{\langle \mathbf{x}, \varepsilon_q(\mathbf{w}) \rangle + \langle \varepsilon_q(\mathbf{x}), \mathbf{w} \rangle - \langle \varepsilon_q(\mathbf{x}), \varepsilon_q(\mathbf{w}) \rangle}_{\varepsilon_q^{D}(\mathbf{x}, \mathbf{w})} + \langle \underbrace{\mathbf{x} - \varepsilon_q(\mathbf{x})}_{q(\mathbf{x})}, \tilde{\varepsilon}_c(\mathbf{w}) \rangle \tag{38}$$

$$- \langle \varepsilon_q(\mathbf{w}), \varepsilon_s(\mathbf{w}) \rangle \tag{39}$$

$$= \varepsilon_{I,s}^{D}(\mathbf{x}, \mathbf{w}) + \varepsilon_q^{D}(\mathbf{x}, \mathbf{w}) + \langle q(\mathbf{x}), \tilde{\varepsilon}_c(\mathbf{w}) \rangle - \langle \varepsilon_q(\mathbf{x}), \varepsilon_s(\mathbf{w}) \rangle \tag{40}$$

After adding the norms, we obtain the following:

$$|\varepsilon_{q,c}^{D}(\mathbf{x}, \mathbf{w})| \leq |\varepsilon_{I,s}^{D}(\mathbf{x}, \mathbf{w})| + |\varepsilon_q^{D}(\mathbf{x}, \mathbf{w})| + |\underbrace{\langle q(\mathbf{x}), \tilde{\varepsilon}_c(\mathbf{w}) \rangle}_{\varepsilon_t}| + |\underbrace{\langle \varepsilon_q(\mathbf{x}), \varepsilon_s(\mathbf{w}) \rangle}_{\varepsilon_i}| \tag{41}$$

where $\varepsilon_t$ and $\varepsilon_i$ are the additional error terms.

To prove non-orthogonality, consider the blocks of floating-point numbers $x = (1.0, 1.0)^T$, $w = (0.6, 1.3)^T$, HBFP4 quantization $q$ and 1:2 sparsity $s$. We assume that $q$ does not affect $x$: $q(x) = x$. On the other hand, the block $w$ is transformed in the following way:

$$s(\mathbf{w}) = \begin{pmatrix} 0 \\ 1.3 \end{pmatrix} \quad q(\mathbf{w}) = \begin{pmatrix} 0.625 \\ 1.25 \end{pmatrix} \quad s(q(\mathbf{w})) = s(q(\mathbf{w})) = c(\mathbf{w}) = \begin{pmatrix} 0 \\ 1.25 \end{pmatrix} \tag{42}$$

The dot product error of the composition equals:

$$\varepsilon_{q,c}^{D}(\mathbf{x}, \mathbf{w}) = \langle \mathbf{x}, \mathbf{w} \rangle - \langle q(\mathbf{x}), c(\mathbf{w}) \rangle = \left\langle \begin{pmatrix} 1.0 \\ 1.0 \end{pmatrix}, \begin{pmatrix} 0.6 \\ 1.3 \end{pmatrix} \right\rangle - \left\langle \begin{pmatrix} 1.0 \\ 1.0 \end{pmatrix}, \begin{pmatrix} 0 \\ 1.25 \end{pmatrix} \right\rangle = 0.65 \tag{43}$$

The dot product error of quantization equals:

$$\varepsilon_q^{D}(\mathbf{x}, \mathbf{w}) = \langle \mathbf{x}, \mathbf{w} \rangle - \langle q(\mathbf{x}), q(\mathbf{w}) \rangle = \left\langle \begin{pmatrix} 1.0 \\ 1.0 \end{pmatrix}, \begin{pmatrix} 0.6 \\ 1.3 \end{pmatrix} \right\rangle - \left\langle \begin{pmatrix} 1.0 \\ 1.0 \end{pmatrix}, \begin{pmatrix} 0.625 \\ 1.250 \end{pmatrix} \right\rangle = 0.025 \tag{44}$$

The dot product error of sparsity equals:

$$\varepsilon_{I,s}^{D}(\mathbf{x}, \mathbf{w}) = \langle \mathbf{x}, \mathbf{w} \rangle - \langle \mathbf{x}, s(\mathbf{w}) \rangle = \left\langle \begin{pmatrix} 1.0 \\ 1.0 \end{pmatrix}, \begin{pmatrix} 0.6 \\ 1.3 \end{pmatrix} \right\rangle - \left\langle \begin{pmatrix} 1.0 \\ 1.0 \end{pmatrix}, \begin{pmatrix} 0 \\ 1.3 \end{pmatrix} \right\rangle = 0.6 \tag{45}$$

Therefore, for these particular values of $\mathbf{x}$ and $\mathbf{w}$, the inequality: $|\varepsilon_{q,c}^{D}(\mathbf{x}, \mathbf{w})| > |\varepsilon_q^{D}(\mathbf{x}, \mathbf{w})| + |\varepsilon_{I,s}^{D}(\mathbf{x}, \mathbf{w})|$ holds true.

$\square$

**Theorem J.2.** Let $q$ be the max-scaled block-wise quantization and $s$ be the magnitude-based N:M sparsity transformation. Then:

$$\forall \mathbf{x} \in \mathbb{R}^n, \quad \|\varepsilon_{q \circ s}(\mathbf{x})\|_1 \leq \|\varepsilon_{s \circ q}(\mathbf{x})\|_1 \tag{46}$$

*Proof.* If sparsity is applied first, then from the proof of Theorem 3.5

$$\varepsilon_{q \circ s}(\mathbf{x})_i = \begin{cases} \varepsilon_s(\mathbf{x})_i & \text{if } x_i \text{ is sparsified,} \\ \varepsilon_q(\mathbf{x})_i & \text{otherwise.} \end{cases} \tag{47}$$

If quantization is applied first, there are two cases.

**Case 1**: there are no new duplicates. Quantization cannot reorder elements, it can only make them duplicates. Therefore, if there are no new duplicates, the same elements will be sparsified after quantization as the order of the elements did not change, and the error vector will be the same:

$$\|\varepsilon_{s \circ q}(\mathbf{x})\|_1 = \|\varepsilon_{q \circ s}(\mathbf{x})\|_1 \tag{48}$$

**Case 2**: there are new duplicates. Let $x_i$ and $x_j$ be such elements that $|x_i| < |x_j|$ and $q(\mathbf{x})_i = q(\mathbf{x})_j =: y$, and the $j$-th element gets sparsified instead of the $i$-th. In this case:

$$\varepsilon_{s \circ q}(\mathbf{x})_i = \varepsilon_q(\mathbf{x})_i \text{ and } \varepsilon_{s \circ q}(\mathbf{x})_j = \varepsilon_s(\mathbf{x})_j = x_j \tag{49}$$

Therefore,

$$|\varepsilon_{soq}(\mathbf{x})_i| + |\varepsilon_{soq}(\mathbf{x})_j| = |\varepsilon_q(\mathbf{x})_i| + |\varepsilon_s(\mathbf{x})_j| = |y - x_i| + |x_j| \tag{50}$$

If we consider the case $x_i < y < x_j$ and $y = 0$, then

$$|\varepsilon_q(x)_i| + |\varepsilon_s(x)_j| = |x_i| + |x_j| = |\varepsilon_s(x)_i| + |\varepsilon_q(x)_j| \tag{51}$$

Otherwise, we assume that $x_i$, $x_j$ and $y$ have the same sign.

Here we have three subcases:

- $|y| > |x_j|$. Then

$$|y - x_i| + |x_j| = |y| - |x_i| + |x_j| > |y| + |x_i| - |x_j| = |y - x_j| + |x_i| \tag{52}$$

- $|y| < |x_i|$. Then

$$|y - x_i| + |x_j| = |x_i| - |y| + |x_j| = |y - x_j| + |x_i| \tag{53}$$

- $|x_i| < |y| < |x_j|$. Then

$$|y - x_i| + |x_j| = |y| - |x_i| + |x_j| > |x_j| = |x_j| - |y| + |y| > |x_j| - |y| + |x_i| = |y - x_j| + |x_i| \tag{54}$$

Therefore,

$$\varepsilon_q(x)_i| + |\varepsilon_s(x)_j| = |y - x_i| + |x_j| \geq |y - x_j| + |x_i| = |\varepsilon_q(x)_j| + |\varepsilon_s(x)_i| \tag{55}$$

As a result,

$$\|\varepsilon_{soq}(x)\|_1 = \left( \sum_{k \neq i,j} |\varepsilon_{soq}(x)_k| \right) + |\varepsilon_{soq}(x)_i| + |\varepsilon_{soq}(x)_j| = \tag{56}$$

$$= \left( \sum_{k \neq i,j} |\varepsilon_{soq}(x)_k| \right) + |\varepsilon_q(x)_i| + |\varepsilon_s(x)_j| \geq \tag{57}$$

$$\geq \left( \sum_{k \neq i,j} |\varepsilon_{qos}(x)_k| \right) + |\varepsilon_q(x)_j| + |\varepsilon_s(x)_i| = \|\varepsilon_{qos}(x)\|_1 . \tag{58}$$

$\square$

## K  DEFINITIONS OF $Q$ FOR NUMERICAL FORMATS

1. INT$m$ (Symmetric version) (Dettmers et al., 2022)

$$Q_m(x_i, scale) = s \cdot \left\lfloor \frac{x_i}{s} \right\rceil, \text{ where } s = \frac{scale}{2^{m-1} - 1}, \tag{59}$$

2. HBFP$m$ (Drumond et al., 2018a)

$$Q_m(x_i, scale) = s \cdot \left\lfloor \frac{x_i}{s} \right\rceil, \text{ where } s = 2^{\lceil log_2(scale) \rceil - (m-1)} \tag{60}$$

3. MXFP$m$ (Darvish Rouhani et al., 2023; Microsoft, 2024)

$$Q_m(x_i, scale) = scale \cdot (-1)^S \cdot 2^E \cdot (1 + 2^{-m} \cdot M) \tag{61}$$

$$\text{where } S = \text{sign}(x_s) \tag{62}$$

$$E = \lfloor log_2(|x_s|) \rfloor - bias \tag{63}$$

$$M = \left\lfloor \left( \frac{|x_s|}{2^E} - 1 \right) \cdot 2^m \right\rceil \tag{64}$$

$$x_s = \frac{x}{2^{\lceil log_2(scale) \rceil}} \tag{65}$$

$$\tag{66}$$

Value of $bias$ depends on the chosen configuration (Micikevicius et al., 2023).

4. MXINT$m$ (Darvish Rouhani et al., 2023; Microsoft, 2024)

$$Q_m(x_i, scale) = s \cdot \left\lfloor \frac{x_i}{s} \right\rceil, \text{ where } s = 2^{\lceil log_2(scale) \rceil - (m-1)} \tag{67}$$

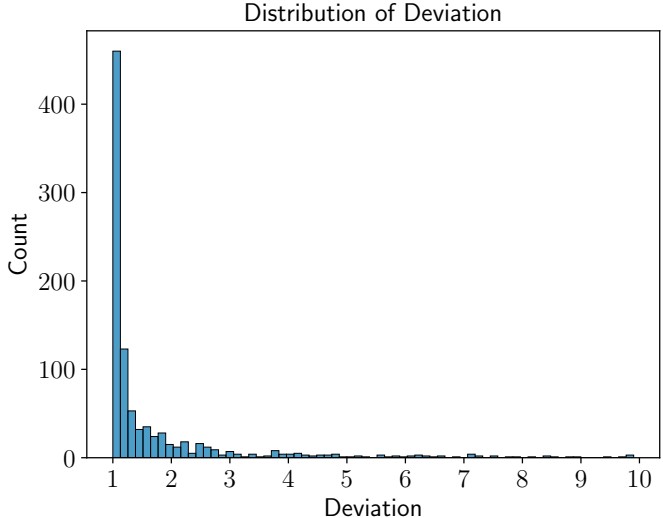

Figure 4: Distribution of the deviation values for several random blocks.

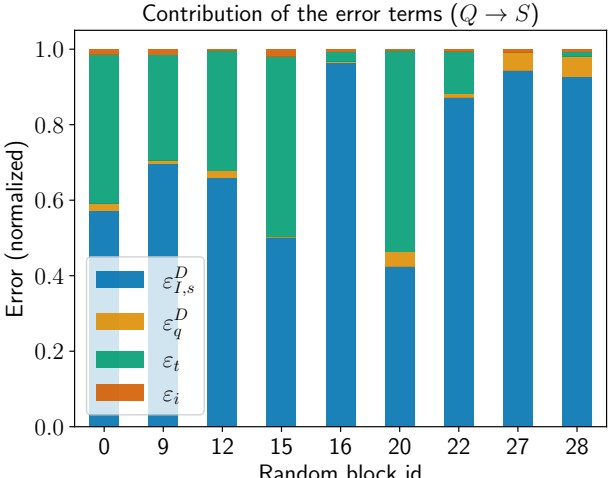

Figure 5: Normalized values of each error term of the upper bound for the case of applying quantization before sparsity.

## L    ANALYSIS OF THE UPPER BOUND OF THE DOT PRODUCT ERROR

### L.1    UPPER BOUND IS REACHABLE

To test if the upper bound derived in Theorem 3.10 is reachable in practice, we randomly sampled 1000 blocks of size 64 from a standard normal distribution $\mathcal{N}(0, 1)$, and applied 2:4 sparsity and HBFP6 quantization. We then compute the aggregate error of the composition and individual error term of the upper bound. Subsequently, we quantified how much the upper bound deviates from the actual composition error using the following formula:

$$\text{Deviation} = \frac{|\varepsilon_{I,s}^D(\mathbf{x}, \mathbf{w})| + |\varepsilon_q^D(\mathbf{x}, \mathbf{w})| + |\varepsilon_t| + |\varepsilon_i|}{|\varepsilon_{q,c}^D(\mathbf{x}, \mathbf{w})|} \tag{68}$$

As a corollary of Theorem 3.10, the minimal value of deviation is 1.

Figure 4 shows the deviation distribution of samples. Most values fall into the first bin, suggesting the upper bound is frequently reached. It can also be seen that the deviation values can be large, almost reaching the value of 10, which indicates that the upper bound can be pessimistic in some cases too. Theorem 3.10 does not rule out

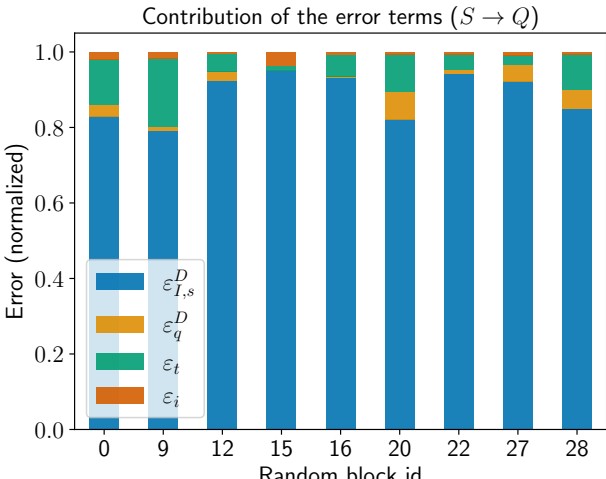

Figure 6: Normalized values of each error term of the upper bound for the case of applying sparsity before quantization. We fix the seed and consider the same random blocks as for the order $Q \to S$.

large values of deviation, as it applies the triangle inequality to obtain the upper bound, which leads to dropping the sign of each error term. If the values of the error terms are negative, they can make the overall error of the composition lower than the upper bound, which leads to the deviation values larger than one.

## L.2 CONTRIBUTION OF ADDITIONAL ERROR TERMS

Section 3.2 describes how each additional error term can contribute to the overall error of the composition. We hypothesize that the error term $\varepsilon_t$ contributes less in case of applying sparsity followed by quantization than in case of applying quantization first. We also hypothesize that the magnitude of the error term $\varepsilon_i$ is much lower than the magnitude of $\varepsilon_t$. To test our hypotheses, we normalized the values of each term of the upper bound to compare their contribution to the error. We considered both orders of applying the transformations. We also only looked at the samples with low deviation values ($< 1.05$) to increase the explainability power of the upper bound.

Figures 5 and 6 depict the results of the experiment. If we consider the order $Q \to S$ in Figure 5, we can see that the term $\varepsilon_t$ advocates for almost half of the error of the composition. However, in the order $S \to Q$ the term $\varepsilon_t$ has a much lower impact, which proves our first hypothesis.

We can also see that the values of $\varepsilon_i$ are much lower that the values of $\varepsilon_t$ in most of the cases in both orders. This proves our second hypothesis.

## M ANALYSIS OF THE ADDITIONAL ERROR

As a corollary of Theorem 3.10, the composition of max-scaled sparsity and quantization is *non-orthogonal*, resulting in two additional error terms.

The term $\varepsilon_t$ incorporates the correction vector of the composition $\tilde{\varepsilon}_c$, which carries the additional error from the tensor level to the dot-product level. Depending on the order of the composition, the value of $\varepsilon_t$ varies.

If sparsity precedes quantization, the correction vector $\tilde{\varepsilon}_{q \circ s}(\mathbf{w})$ exclusively comprises negative quantization errors for the elements pruned by the sparsity transformation:

$$\tilde{\varepsilon}_{q \circ s}(\mathbf{w})_i = \begin{cases} -\varepsilon_q(\mathbf{w})_i, & s(\mathbf{w})_i = 0 \\ 0, & \text{otherwise} \end{cases} \tag{69}$$

However, if quantization is applied first, certain elements in the block may become equal, resulting in the sparsity removing a different set of elements. Formally, if $w_i$ is pruned by $s$ but not by $s \circ q$, there exists a $w_j$, where $j \neq i$, such that $q(\mathbf{w})_i = q(\mathbf{w})_j$. In this scenario, $\tilde{\varepsilon}_{s \circ q}(\mathbf{w})_i = -\varepsilon_s(\mathbf{w})_i$ and $\tilde{\varepsilon}_{s \circ q}(\mathbf{w})_j = \varepsilon_s(\mathbf{w})_j - \varepsilon_q(\mathbf{w})_j$. Otherwise, it only contains the quantization errors of the pruned elements. Therefore, the magnitude of the correction vector for the composition $s \circ q$ is generally larger than that of the reverse order. Besides the quantization errors, the correction vector also contains pruned elements, which are generally larger by orders of

magnitude except in a few improbable edge cases. This results in the overall value of $\varepsilon_t$ being larger, implying that *the order of applying sparsity first and then quantization is optimal for dot products.*

The term $\varepsilon_i$ also contributes to the additional error, encoding the interaction between the error vectors $\varepsilon_q(\mathbf{x})$ and $\varepsilon_s(\mathbf{w})$. However, $\varepsilon_i$ is less significant than $\varepsilon_t$, as it contains the quantization error, the norm of which is generally orders of magnitudes lower than the norm of the original block. In addition, $\varepsilon_i$ contains the sparsity error, which involves the smallest weights, diminishing the significance of this additional error.

## N   COMBINING GPTQ WITH SPARSITY

**Magnitude-based Sparsity and GPTQ.** Although our mathematical analysis does not specifically cover GPTQ, we conducted controlled experiments to evaluate its performance in the context of our paper. We applied 50% unstructured sparsity to the OPT-125M model in combination with GPTQ.

Table 10: GPTQ - Magnitude Based for OPT-125m

| S&Q Layer Id | Sparsity | Quantization | Order | PPL |
|---|---|---|---|---|
| - | - | - | - | 27.65 |
| 0, 1, 10, 11 | - | GPTQ-4b | - | 28.1 |
| 0, 1, 10, 11 | 50% | - | - | 34.62 |
| 0, 1, 10, 11 | 50% | GPTQ-4b | S $\rightarrow$ Q | 30.93 |
| 0, 1, 10, 11 | 50% | GPTQ-4b | Q $\rightarrow$ S | 35.59 |

Table 11: GPTQ - SparseGPT

| Sparsity type | Order | OPT-350M | | | OPT-1.3B | | |
|---|---|---|---|---|---|---|---|
| | | Wikitext2 | PTB | C4 | Wikitext2 | PTB | C4 |
| 2:4 | S $\rightarrow$ Q | 56.27 | 80.67 | 51.65 | 27.99 | 42.32 | 29.42 |
| | Q $\rightarrow$ S | 67.96 | 91.03 | 56.57 | 29.38 | 45.15 | 30.15 |
| 3:4 | S $\rightarrow$ Q | 26.82 | 38.45 | 26.86 | 16.73 | 24.30 | 18.62 |
| | Q $\rightarrow$ S | 27.38 | 39.62 | 27.21 | 17.42 | 24.23 | 18.72 |
| 4:8 | S $\rightarrow$ Q | 43.06 | 62.84 | 39.09 | 22.96 | 33.61 | 23.86 |
| | Q $\rightarrow$ S | 46.49 | 64.53 | 42.05 | 24.75 | 35.88 | 25.40 |

When applying GPTQ$\rightarrow$S, finetuning requires quantizing and sparsifying weight tensors in tandem at each iteration. However, GPTQ operates by quantizing a column and updating the remaining weights to compensate for the introduced errors. Under this scenario, comparing GPTQ$\rightarrow$S and S$\rightarrow$GPTQ with finetuning would not be fair due to the different amounts of error compensation. To ensure a fair comparison, we decided to eliminate the fine-tuning step and instead sparsified only a subset of layers to contain the sparsity error to a reasonable degree. We determined the number of layers to compress by setting a perplexity threshold equivalent to that achieved by SparseGPT. Table 10 shows that even in this case, magnitude-based sparsity is most effective when applied before quantization (S$\rightarrow$GPTQ: 30.93 vs. GPTQ$\rightarrow$S: 35.59).

**SparseGPT and GPTQ.** GPTQ and SparseGPT apply the compression on a column-by-column basis and assume that elements to the right of the current column remain uncompressed. This is because dense updates propagate through these uncompressed elements to compensate for the introduced error. If subsequent columns are compressed, they would not remain so after the first update.

Given this context, we used the SparseGPT codebase, which natively supports S$\rightarrow$Q, and followed their instructions to apply this compression order. We also reached out to the author of SparseGPT and followed their recommendation to apply Q$\rightarrow$S. We experimented with 4-bit GPT quantization and different variants of OPT models. Table 11 summarizes our results, supporting our hypothesis for the optimal order of compression for second-order methods. The mathematical study of the optimal compression order in second-order methods is beyond the scope of our work, and we leave it as future work.

## O   CONVOLUTIONAL NETWORKS

Our mathematical framework is designed to be applicable to any matrix multiplications, regardless of the specific model architecture. This allows us to study the optimal order of compression for various models, including CNNs. Therefore, we extended our experiments to include ResNet50 on the ImageNet dataset using all of the same configurations as ViT. The results are present in Table 12. These additional results further validate our findings regarding the optimal compression order and orthogonality threshold.

## P   OPTIMAL ORDER WITHOUT FINE-TUNING

Magnitude-based sparsity, when applied without further re-training, leads to significant accuracy degradation (Hoefler et al., 2021; Frantar & Alistarh, 2023). In our case, without sparsity-aware fine-tuning, the sparsity error becomes several orders of magnitude larger than the quantization error, causing both S$\rightarrow$Q and Q$\rightarrow$S to yield predominantly sparsity error. Table 13 shows the WikiText2 perplexities of FP32 sparse models without sparsity-aware finetuning.

Table 12: Comparison of evaluation cross-entropy loss with estimated orthogonality thresholds

| Sparsity type | Number format | ResNet50 | | | |
| | | Metric | | Orthogonality Threshold | |
| | | Accuracy | CE Loss | Accuracy | CE Loss |
|---|---|---|---|---|---|
| 0% | FP32 | 76.97% | 1.040 | - | - |
| | INT8 | 76.83% | 1.051 | - | - |
| | MXFP8 | 69.21% | 1.462 | - | - |
| | MXFP6 | 70.86% | 1.362 | - | - |
| | HBFP8 | 76.88% | 1.043 | - | - |
| | HBFP6 | 74.61% | 1.176 | - | - |
| 50% | FP32 | 76.33% | 1.067 | - | - |
| | INT8 | 76.06% | **1.072** | **76.19%** | 1.078 |
| | MXFP8 | 62.07% | 1.917 | **68.57%** | **1.489** |
| | MXFP6 | 69.54% | 1.420 | **70.22%** | **1.389** |
| | HBFP8 | 76.21% | 1.072 | **76.24%** | **1.070** |
| | HBFP6 | 73.95% | **1.186** | **73.97%** | 1.204 |
| 2:4 | FP32 | 76.90% | 1.044 | - | - |
| | INT8 | 76.49% | 1.060 | **76.66%** | **1.055** |
| | MXFP8 | 67.03% | 1.580 | **69.04%** | **1.467** |
| | MXFP6 | 70.47% | **1.351** | **70.69%** | 1.366 |
| | HBFP8 | 76.51% | 1.053 | **76.71%** | **1.047** |
| | HBFP6 | **74.51%** | **1.158** | 74.44% | 1.181 |
| 75% | FP32 | 67.30% | 1.569 | - | - |
| | INT8 | **67.33%** | **1.565** | 67.16% | 1.580 |
| | MXFP8 | **62.14%** | **1.920** | 59.54% | 1.991 |
| | MXFP6 | 58.77% | 2.785 | **61.19%** | **1.891** |
| | HBFP8 | 67.13% | 1.581 | **67.21%** | **1.572** |
| | HBFP6 | 64.78% | 1.733 | **64.94%** | **1.706** |
| 1:4 | FP32 | 73.91% | 1.218 | - | - |
| | INT8 | **73.89%** | **1.227** | 73.77% | 1.229 |
| | MXFP8 | 58.11% | 2.314 | **66.15%** | **1.640** |
| | MXFP6 | 63.80% | 1.907 | **67.80%** | **1.540** |
| | HBFP8 | 73.79% | 1.225 | **73.82%** | **1.221** |
| | HBFP6 | 71.47% | **1.340** | **71.55%** | 1.355 |

Table 13: Magnitude-based sparsity applied without fine-tuning

| Sparsity | OPT-125m | OPT-6.7b | LLaMA-2-7B | LLaMA-3-8b |
|---|---|---|---|---|
| 50% | 146. | 81. | 27. | 34. |
| 2:4 | 619. | 238. | 76. | 113. |

The significant degradation in accuracy makes the scenario without sparsity-aware finetuning impractical. Therefore, we reproduced the configurations from Table 1 with only a portion of the layers being sparse and quantized. By applying compression to approximately one-third of the layers, specifically those located at the beginning and end of the model, we can achieve acceptable perplexity increases. These results support the optimality of the S→Q order.

## Q  COLLISION ANALYSIS

In the proof of Theorem J.1, we demonstrated that additional error may arise in sub-optimal order when, after quantization, a larger value and a smaller value collide, causing the originally larger element to be pruned. In this section, we provide empirical evidence of the fact that these collisions take place.

We extracted the weights of all layers of the OPT-1.3b model and calculated the average reduction in unique elements after applying quantization (HBFP6). The reduction in unique elements for each layer is computed as:

$$\Delta unique = unique(W) - unique(\hat{W}) \tag{70}$$

where $unique(X)$ returns the number of unique elements in $X$. On average, $\Delta unique(W, \hat{W}) = 24230$, with the total number of unique elements before quantization averaging $24419$. Therefore, quantization introduces significant collisions, which may lead to additional error.

Table 14: Perplexities of OPT-125m model without sparsity-aware fine-tuning

| S&Q Layer Id | Sparsity type | Order | INT8 | MXFP8 | MXFP6 | HBFP8 | HBFP6 | HBFP4 |
|---|---|---|---|---|---|---|---|---|
| | | | **OPT-125M** | | | | | |
| 0. 1, 10, 11 | 50% | S→Q | **35.85** | **34.98** | **35.02** | **34.73** | **35.41** | **230.** |
| | | Q→S | 35.92 | 35.03 | 35.23 | 34.97 | 36.05 | 305. |
| | 2:4 | S→Q | **42.97** | **40.5** | **40.51** | **40.19** | **43.43** | **471.** |
| | | Q→S | 43.32 | 40.61 | 40.59 | 40.91 | 48.64 | 793. |
| | | | **LLaMA-2-7B** | | | | | |
| 0-4, 27-31 | 50% | S→Q | **8.69** | **8.73** | **10.04** | **8.72** | **9.98** | **18.** |
| | | Q→S | 8.85 | 8.74 | 11.44 | 8.68 | 10.43 | 26. |
| | 2:4 | S→Q | **9.31** | **9.39** | **11.21** | **9.25** | **10.06** | **31.** |
| | | Q→S | 9.61 | 9.92 | 11.67 | 9.70 | 11.24 | 44. |

To further validate our findings, we analyzed the reduction of unique elements at the block level rather than across the entire tensor. Specifically, we examined the 6th layer of OPT-1.3B, selecting random blocks with a block size of 64. As shown in Figure 7, the reduction of unique elements in a block can reach up to 41, representing approximately 64% of the elements in the block. These results confirm that quantization introduces a substantial number of duplicates.

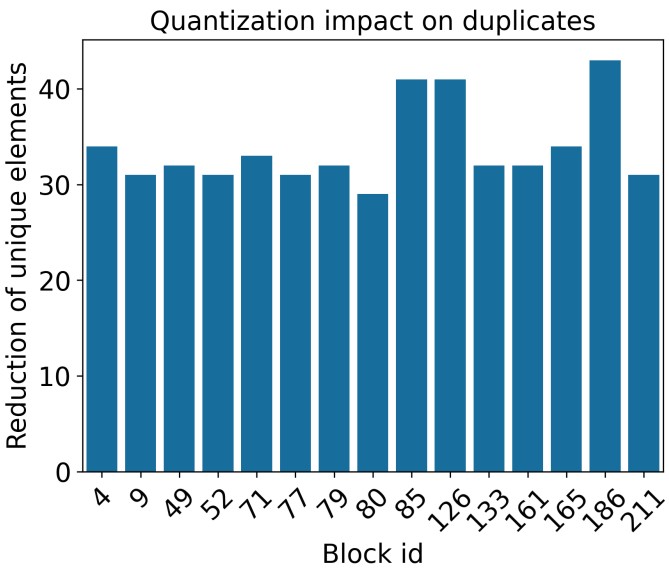

Figure 7: Impact of quantization the number of unique elements in each block.

## R OPENLLM EVALUATION

To further validate our findings and expand our experimental setup, we evaluated the models on zero-shot tasks Gao et al. (2024) under both compression orders (S→Q and Q→S). The results are presented in Table 15 for OPT-125M and Table 16 for Llama-3-8B. The results show that in all cases, the optimal compression order (S→Q) yields better performance across all models and tasks, validating our theoretical conclusions.

## S ORDER OF SPARSITY AND QUANTIZATION DURING FINE-TUNING

Our mathematical analysis of sparsity and quantization assumes the model weights to be fixed. However, in our empirical study, the master weights of compressed layers change during fine-tuning. To bridge the gap between the static case in our mathematical analysis and the dynamic case in our experimental setup, we demonstrate

that the optimal compression order is independent of the exact model weights, and that quantization minimally affects the model weights during sparsity-aware fine-tuning.

**Zero-shot sparsity and quantization.** To show that the optimal compression order is independent from the model weights, we apply zero-shot compression to intermediate dense FP32 checkpoints of the OPT-125M model on WikiText2. Table 17 demonstrates final perplexities after applying HBFP6 quantization and 50% unstructured sparsity in S→Q and Q→S orders at various checkpoints of dense fine-tuning. The perplexity increases as dense fine-tuning progresses, which is expected, as the weights become less tolerant to sparsity during the dense FP32 fine-tuning process. For each checkpoint, S→Q consistently outperforms Q→S, achieving a relative gap of up to 7%. This confirms that the optimal compression order is independent of the model's specific weights, validating our mathematical analysis.

**Weight distributions during fine-tuning.** In our experiments, we employ sparsity-aware fine-tuning to mitigate sparsity-induced errors. A detailed summary of our experimental set-up is presented in Table 6. During fine-tuning, we store master weights in full precision for each layer, following prior work (Rouhani et al., 2023b). This approach enables compression of matrix multiplications during the forward and backward stages while retaining full precision for weight updates. Consequently, quantization during forward and backward phases has minimal impact on the learning dynamics of the weights.

To demonstrate the minimal effect of quantization during fine-tuning, we compare the distributions of master weights at various iterations during S→Q and Q→S fine-tuning. Figures 8, 9, 10 show that the master weights remain nearly identical between the two schedules, with negligible differences in the small magnitude range $[-0.2, 0.2]$. Although weight values in the tails of the distributions exhibit more noticeable discrepancies, reordering sparsity and quantization impacts only the smallest values within each block, which lie within the range where the distributions are nearly identical.

To further quantify these differences, Figure 11 presents the distribution of discrepancies for a single weight component. The absolute difference between weight values remains below 0.02, which is negligibly small at the tensor and model levels. Therefore, mathematical analysis remains valid even during fine-tuning.

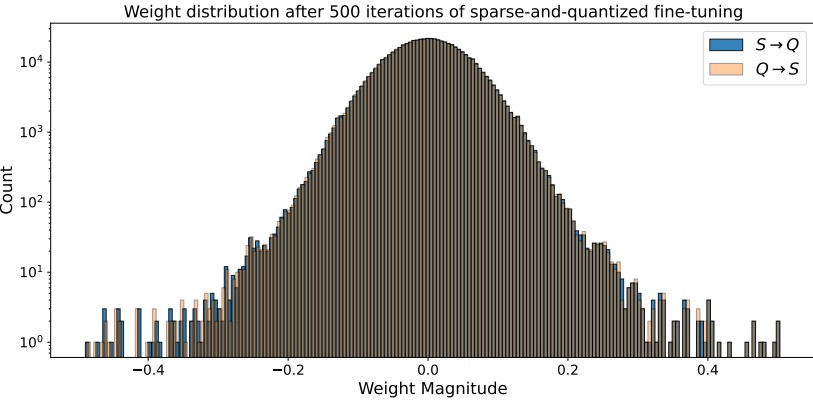

Figure 8: Distribution of master weights of Q_proj layer at the beginning of fine-tuning

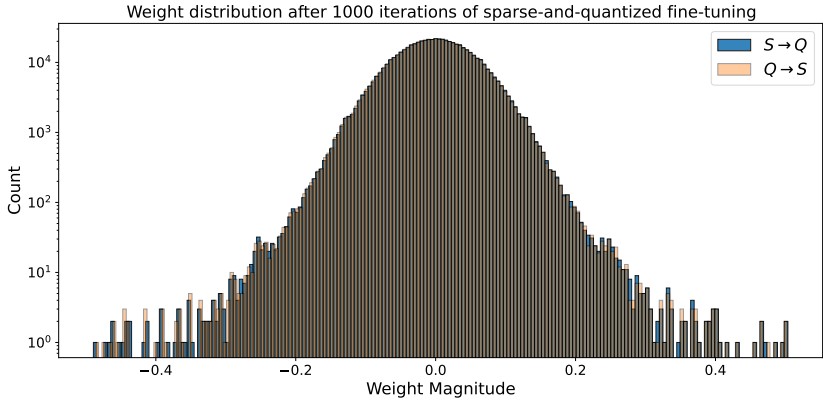

Figure 9: Distribution of master weights of Q_proj layer at the middle of fine-tuning

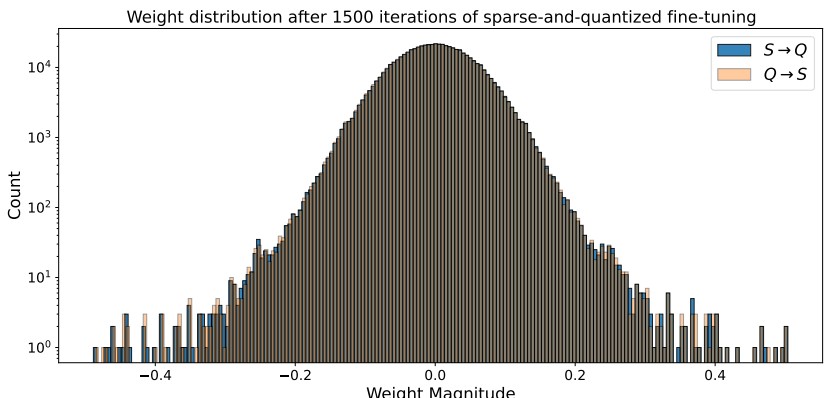

Figure 10: Distribution of master weights of Q_proj layer at the end of fine-tuning

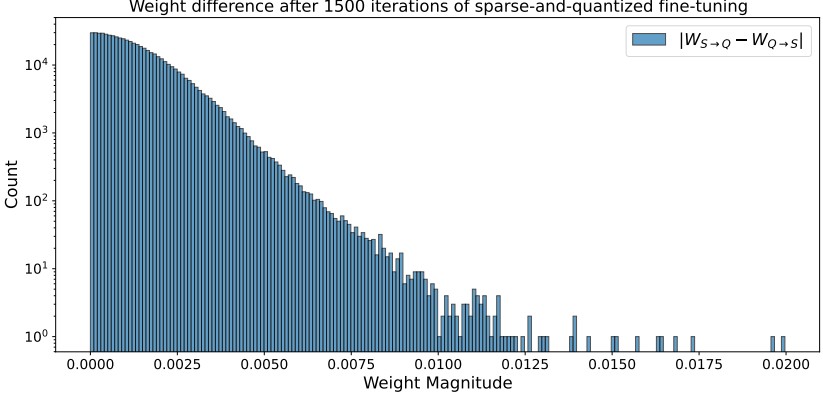

Figure 11: Distribution of discrepancies between weights of Q_proj layer at the end of fine-tuning

Table 15: OPT-125M Zero-Shot Performance. The best results for each configuration are highlighted in bold.

| Sparsity | Num format | Order | ARC-c | ARC-e | HellaSWAG | WinoGrande |
|----------|-----------|-------|-------|-------|-----------|------------|
| 0% | HBFP6 | - | 19.62 | 41.79 | 28.74 | 51.46 |
| | INT8 | - | 19.03 | 36.28 | 28.45 | 50.99 |
| 50% | HBFP6 | S→Q | **20.31** | **39.86** | **28.01** | **51.38** |
| | | Q→S | 19.04 | 38.56 | 26.41 | 50.67 |
| | INT8 | S→Q | **20.99** | **34.18** | **27.49** | **52.88** |
| | | Q→S | 20.12 | 33.70 | 26.98 | 51.14 |
| 2:4 | HBFP6 | S→Q | **20.31** | **38.26** | **27.56** | **52.17** |
| | | Q→S | 18.92 | 36.68 | 27.23 | 49.96 |
| | INT8 | S→Q | **19.97** | **34.93** | **27.81** | **50.67** |
| | | Q→S | 18.51 | 30.17 | 27.35 | 48.74 |

Table 16: LLaMA-3 Zero-Shot Performance. The best results for each configuration are highlighted in bold.

| Sparsity | Num format | Order | ARC-c | ARC-e | HellaSWAG | WinoGrande |
|----------|-----------|-------|-------|-------|-----------|------------|
| 0% | HBFP6 | - | 48.21 | 76.43 | 59.17 | 71.19 |
| 50% | HBFP6 | S→Q | **37.54** | **69.19** | **50.64** | **64.25** |
|  |  | Q→S | 37.29 | 67.59 | 49.54 | 63.38 |
| 2:4 | HBFP6 | S→Q | **31.91** | **59.6** | **45.85** | **60.77** |
|  |  | Q→S | 29.83 | 59.48 | 42.9 | 58.43 |

Table 17: Perplexities of the intermediate checkpoints during dense fine-tuning of OPT-125m, HBFP6+50% sparsity. The best results for each configuration are highlighted in bold.

| Steps | 500 | 700 | 900 | 1100 | 1300 | 1500 | 1700 |
|---|---|---|---|---|---|---|---|
| S→Q | **86.09** | **95.61** | **100.35** | **97.22** | **119.66** | **126.04** | **130.31** |
| Q→S | 89.03 | 97.91 | 107.28 | 98.58 | 125.10 | 127.35 | 131.01 |

