# OpenReview forum: "Effective Interplay between Sparsity and Quantization: From Theory to Practice"
_ICLR.cc/2025/Conference — ICLR 2025 Spotlight_

### Official Review · Reviewer_e8N1 · 2024-11-01

**Soundness:** 2
**Presentation:** 3
**Contribution:** 3
**Rating:** 6
**Confidence:** 4

**Summary:**

The authors present a theoretical proof showing that the order in which magnitude-based pruning and scaled block quantization are performed is of importance in the context of preserving model performance. The authors define notions such as orthogonality of two operations — which is when the composition of the two operations does not result in any additional error than applying each individual transformation. The authors show theoretically that magnitude pruning and quantization are not orthogonal operations (when going beyond tensor-level) and further showed, both theoretically and empirically, that applying pruning first and then quantization generally leads to better performance.

**Strengths:**

- Notation, definitions, and theorems in Section 3 are generally clear and their significance is adequately articulated.
- The authors have addressed an issue that has gone overlooked in the pruning/quantization literature through both theoretical proofs and derivations as well as empirical studies that further solidify their claims.

**Weaknesses:**

- The discussion following Theorem 3.9 is very hard to digest for a reader who hasn’t spent as much time as the authors thinking about this problem. I’d encourage the authors to prune the text, retaining only the essential message (which presumably is what’s written in italics) and moving other information to the Appendix.
- Overall, the theoretical claims and experiments are not astonishing as one would perhaps expect that pruning should precede quantization.
- The theoretical contribution is quite limited as it only holds for magnitude-based pruning (without fine tuning) and block-wise quantization. Importantly, magnitude pruning has gone out of fashion in the context of LLMs because it requires costly fine-tuning to recover model performance and is outperformed by methods like SparseGPT and WANDA when fine-tuning is not performed. The authors mention in the Appendix that, empirically, the order had less of an impact for WANDA and SparseGPT.
- The experiments seem to be quite orthogonal to the theoretical results. By employing fine-tuning for all the experiments, the authors are making their original theoretical proofs/derivations inapplicable in the context of the experiments as the derivations are based on errors calculated when no fine-tuning is applied.
- Proof of Theorem 3.5: Only show equality is attained for L1 norm and not all norms. Is it clear that this implies that equality is also achieved for all other norms? Statement of Theorem or proof should be modified to address this.
- Proof of Theorem 3.6 is only a counter-example for the L1 norm. Is it immediate that the theorem is true in general for norms beyond the L1 norm? Either the statement of the theorem or the proof should be modified to address this.
- Throughout the paper, some statements are true for all norms, others are only shown for the L1 norm, and then the empirical experiments utilize the L2 norm for measuring errors.
- The generalization of orthogonality in Definition 3.8 is not clear to me as functions are now being applied coordinate-wise. Is the composition only permitted to happen in one coordinate (similar to in Theorem 3.9). It might be worth it to explicitly write out the definition as the lack of an explicit definition of orthogonality also makes the statement of Theorem 3.9 confusing.

**Questions:**

- In Section 3, the authors mention performing quantization at the level of “blocks." Could you clarify what you mean by a “block” in this context? Does it refer to a set of weights associated with a CNN filter, or does it resemble the M:N sparsity blocks? Or is it something entirely different?
- Consider renaming Definition 3.4 to avoid confusion, as it defines "orthogonality" between two functions in a way that diverges from the standard interpretation. Traditionally, orthogonality between functions is defined by the condition \(\int f(x) g(x) \, dx = 0\), so using "orthogonality" here might lead to misinterpretation.
- Why is it important to consider block-wise quantization in Section 3? Since it’s a theoretical derivation, why don’t you simply assume quantization on the tensor level?
- Theorem 3.5 assumes “max-scaled block-wise quantization”. Is such quantization prevalent in the literature and in practice?
- Theorems 3.5 and 3.6 imply that the optimal order is pruning followed by quantization. Theorem 3.7 analyses the error for the suboptimal order. Why is that of interest?
- Is Equation 12 a lower bound or an upper bound? You might want to rename it accordingly to “Orthogonality Lower Bound” or “Orthogonality Upper Bound” to help the reader.
- “If the compression methods are non-orthogonal, and the evaluation metric indicates better model performance with lower values, we expect the compressed model’s evaluation metric to exceed the orthogonality bound.” — I read this sentence several times and I still can’t understand it. What do you mean by “lower values”? Which values?
- “For OPT, LLaMA, ViT, and ResNet fine-tuning, we employ sparse fine-tuning on a dense” — what method exactly are you using? Please cite the paper.
- “we apply one-shot quantization to sparse fine-tuned models” — again, what method exactly are you using? Please cite the paper. is it the “max-scaled block-wise quantization”?
- “we directly fine-tune the model in a quantized and sparsified manner” — how does one fine tune a quantized model? Isn’t there an issue with doing that?
- Could you summarize the “Experimental setup” in the form of a table. Otherwise, there are too many details in the paragraph and it’s very hard to digest.
- In Figure 1, the error accumulates across layers. This stands in contrast to Figure 1 in [1] which shows attenuation of noise injected in intermediate layers. Could it be that the authors should compute a relative error instead of an absolute error (see caption in Figure 1 of that paper)?
- “and/or reduce quantization effective bitwidth.” — what do you mean by “effective bit width”?
- “TOPS/mm2” — what’s TOPS and mm^2?

[1] Stronger Generalization Bounds for Deep Nets via a Compression Approach

---

> ### Author Response · Authors · 2024-11-19
> **[Part 1/5] Clarifications on Theorems, Compression Methods, and Experimental Setup**
>
> Dear Reviewer e8N1,
>
> We appreciate your comments and positive assessment of our work. We believe all your questions can be addressed within this discussion period. Please find our responses below, and feel free to ask further questions if anything remains unclear.
>
> ---
>
> ## W1: Prune the discussion following Theorem 3.9 (now Theorem 3.10)
>
> Thank you for your suggestion. We simplified the text in our revised paper (L286,  Page 6).
>
> Theorem 3.9 is now Theorem 3.10 in the revised manuscript.
>
> ---
>
> ## W2: One would expect that pruning should precede quantization
> We appreciate the reviewer’s observation. While this point is valid, our work extends beyond it by providing mathematical proof and uncovering additional, less obvious factors that contribute to compression error, particularly in the suboptimal Q$\rightarrow$S order.
>
>
> **(a) Clarifying and Communicating Insights:** We agree that intuition is valuable, but it can vary across researchers. A formal proof provides an objective and universal explanation that ensures the validity of the insight is communicated effectively to all researchers, regardless of their initial assumptions. By mathematically analyzing these compression methods, we aim to establish a shared foundation for understanding their interplay.
>
> **(b) Optimal Order of Compression is Not Universally Acknowledged.** Although the intuition for S$\rightarrow$Q exists, at least two prior papers [1,2] have applied these techniques in a suboptimal order (Q $\rightarrow$ S). Additionally, Wu et al. [3] raised the question of optimal compression order, opting for S $\rightarrow$ Q after experiments comparing both orders, but without providing mathematical proof justifying their decision.
>
> Our contribution lies in rigorously addressing this open question by analyzing the underlying error mechanisms in both orders. This systematic approach not only solidifies the understanding of S$\rightarrow$Q as the optimal order but also clarifies why Q$\rightarrow$S introduces compounded errors.
>
> **(c) Demystifying Factors Contributing to Compression Error:** While zero values have no quantization error, our analysis shows that the Q$\rightarrow$S order introduces further error due to local reordering of tensor elements. Specifically, quantization can change the relative ordering of tensor elements, leading to significant elements being incorrectly removed during subsequent pruning. This reordering effect, which we detail in Appendix J (L1338-1340, Page 25), contributes to increased compression error in the suboptimal order and demonstrates why the S$\rightarrow$Q order is optimal.
>
> [1] Benjamin Hawks, Javier M. Duarte, Nicholas J. Fraser, Alessandro Pappalardo, Nhan Tran, Yaman Umuroglu. Ps and Qs: Quantization-Aware Pruning for Efficient Low Latency Neural Network Inference. Frontiers in AI 2021
>
> [2] Peng Hu, Xi Peng, Hongyuan Zhu, Mohamed M. Sabry Aly, Jie Lin. OPQ: Compressing Deep Neural Networks with One-shot Pruning-Quantization. AAAI 2021
>
> [3] Xiaoxia Wu, Cheng Li, Reza Yazdani Aminabadi, Zhewei Yao, Yuxiong He. Understanding Int4 Quantization for Language Models: Latency Speedup, Composability, and Failure Cases. ICML 2023

---

> ### Author Response · Authors · 2024-11-19
> **[Part 2/5] Clarifications on Theorems, Compression Methods, and Experimental Setup**
>
> ## W3: Magnitude-based pruning is outdated, and SparseGPT and Wanda are better
>
> We agree that SparseGPT and Wanda are popular and more computationally efficient than magnitude-based pruning. However, it is also known that these sparsity techniques exhibit accuracy loss even at moderate sparsity levels [1], beyond which they still require fine-tuning. Moreover, SparseGPT and Wanda have not been shown to be effective at sparsity levels beyond 50% [4,5]. The latter is a fundamental trade-off between these techniques and magnitude-based pruning, which is versatile and can achieve better performance across a wider range of sparsity levels, albeit more computationally intensive.
>
> **Prevalence of magnitude-based pruning and block-wise quantization:** Magnitude-based pruning with fine-tuning continues to be adopted in industry and academia because of its ability to preserve quality [2,3]. For models requiring higher sparsity levels or greater accuracy preservation, magnitude pruning often outperforms one-shot methods. In addition, max-scaled block-wise quantization is widely used in both academic literature and industrial applications (please refer to our detailed response to Q4), further justifying our emphasis on these methods.
>
> **Order-Dependence and Pruning Criteria Differences:** The pruning criteria used by SparseGPT and Wanda differ fundamentally from magnitude-based pruning, likely explaining why the order of operations has less impact in these methods (as noted in Appendix F, Page 20). While our current work focuses on magnitude-based pruning to establish foundational insights into the interplay between sparsity and quantization, we strongly agree that extending the mathematical analysis to Hessian-based methods like SparseGPT and Wanda would be a valuable direction for future work.
>
>
> [1] Xudong Lu, Aojun Zhou, Yuhui Xu, Renrui Zhang, Peng Gao, Hongsheng Li. SPP: Sparsity-Preserved Parameter-Efficient Fine-Tuning for Large Language Models. ICML 2024.
>
> [2] https://pytorch.org/blog/accelerating-neural-network-training/
>
> [3] Elias Frantar, Carlos Riquelme Ruiz, Neil Houlsby, Dan Alistarh, Utku Evci. Scaling Laws for Sparsely-Connected Foundation Models. ICLR 2024.
>
> [4] Elias Frantar, Dan Alistarh. SparseGPT: Massive Language Models Can be Accurately Pruned in One-Shot. ICML 2023.
>
> [5] Mingjie Sun, Zhuang Liu, Anna Bair, J. Zico Kolter. A Simple and Effective Pruning Approach for Large Language Models. ICLR 2024.
>
> ---
>
>
> ## W4: Theory and experiments are orthogonal because experiments have fine-tuning
>
> Thank you for your observation. We acknowledge that our mathematical analysis assumes a model with fixed weights, which differs from the dynamic nature of fine-tuning. To address this, we have included zero-shot compression results for both orders in Appendix S (Page 34), providing additional empirical validation for our theoretical analysis. However, we note that our analysis in Sections 3-4 still remains valid. Please see the detailed explanation below:
>
> - Sparsity first, then quantization remains optimal even if the model weights keep changing. To demonstrate this, we applied zero-shot quantization and sparsity to intermediate checkpoints of the dense fine-tuned model and compared perplexities (Table 17, Page 38). Please note that we do not re-train models after applying compression, which matches our mathematical formulation.
>
> - Quantization introduced during fine-tuning has a negligible effect on the learning dynamics of weights. We validate this by analyzing weight distributions at the beginning, middle, and end of sparsity-aware fine-tuning for both orders. The distributions are nearly identical, particularly in the range of near-zero values, which are the most affected by compression (Figures 9, 10, 11, Page 35).
>
> These findings reinforce the validity of our theoretical framework, even under the dynamic conditions of fine-tuning.
>
> ---
>
> ## W5-W6-W7: All the theorems for all the norms
>
> Per your suggestion, we made the following changes:
> - In Definition 3.4 (Tensor-level orthogonality, Page 4), we explicitly stated that orthogonality is defined for $L_p$ norms, where $p \in [1, +\infty)$.
> - In the proofs of Theorem 3.5 and Theorem 3.6, all statements, examples, and counterexamples were revised to be valid for all $L_p$ norms under consideration (Appendix J, Pages 24-25).
> - We added a general formulation of Theorem 3.7 that accounts for all $L_p$ norms (Theorem J.1, Appendix J, page 25).
>
> ---
>
>
> ## W8: The generalization of orthogonality
>
> To clarify the definition of orthogonality on the dot-product level, we added a formal definition to the revised version of the paper (Definition 3.9, Page 5). The composition is applied only to the second coordinate, as sparsity and quantization are applied only to the weights, while activations are only quantized.

---

> ### Author Response · Authors · 2024-11-19
> **[Part 3/5] Clarifications on Theorems, Compression Methods, and Experimental Setup**
>
> ## Q1,3,4: Clarifying Block-Wise Quantization
>
> We have updated the Related Work section to better highlight the prevalence and practical relevance of max-scaled block-wise quantization. Thank you for bringing this to our attention.
>
> Please find a more detailed response below.
>
> **(1) Prevalence of Max-Scaled Block-Wise Quantization (Q4):** Max-scaled block-wise quantization is widely used in both research and industry due to its ability to reduce memory footprint, improve arithmetic density (particularly in sub-8-bit regimes), and preserve accuracy with minimal hardware overhead. These attributes make max-scaled block-wise quantization highly suitable for modern AI accelerators, as demonstrated in prior work by Drumond et al. [5, 6], Zhang et al. [7], and Rouhani et al. [3, 4, 8]. Additionally, a consortium of vendors under the umbrella of the Open Compute Project [2, 3, 4] has proposed block-wise scaled numerical formats based on shared microexponents (MX), enabling two levels of fine-grained scaling on a block of mantissas. Modern hardware platforms have also adopted these techniques; for instance, the upcoming NVIDIA Blackwell GPUs [1] will support MXFP [3], a max-scaled block-wise format optimized for training and inference.
>
> **(2) Definition and importance of a “Block” (Q1):** In Section 3, "blocks" refer to contiguous subsets of tensor elements that share a scaling factor during quantization, as used in block-wise scaling formats [3, 4, 5, 8]. This differs from concepts like CNN filter weights or N:M sparsity blocks. Unlike element-wise formats such as FP32, BFLOAT16, and FP16—where each element has an independent scaling factor—block-wise quantization assigns a shared scaling factor to groups of elements, with block sizes varying by format. For further explanation, please refer to Section 2 of the revised paper, where details from the Appendix have been incorporated into the Related Work section (as suggested by Reviewer KXV4). We believe this change will clarify the notion of "blocks" for readers.
>
> **(3) Block level vs. Tensor level (Q3):** Our paper specifically focuses on block-wise quantization methods where the matrix is divided into independent blocks, each quantized separately. Best practices from the literature [3, 4, 5, 8] use block sizes smaller than the entire tensor (e.g., 32) as they minimize the accuracy loss due to large scaling factors. These block sizes are amenable to hardware acceleration. Because these blocks are quantized independently, quantizing a tensor is equivalent to quantizing N blocks. We analyze blocks in our definitions and theorems because they are the smallest quantization units and naturally extend to the tensor level. The cumulative error of a tensor is the sum of errors across its constituent blocks, making block-level analysis indicative of overall tensor behavior.
>
> [1] https://www.nvidia.com/en-us/data-center/tensor-cores/#blackwell
>
> [2] https://www.opencompute.org/blog/amd-arm-intel-meta-microsoft-nvidia-and-qualcomm-standardize-next-generation-narrow-precision-data-formats-for-ai
>
> [3] Bita Darvish Rouhani et al. Microscaling Data Formats for Deep Learning. arXiv 2023.
>
> [4] Bita Darvish Rouhani et al. With Shared Microexponents, A Little Shifting Goes a Long Way. ISCA 2023.
>
> [5] Mario Drumond, Tao Lin, Martin Jaggi, Babak Falsafi: Training DNNs with Hybrid Block Floating Point. NeurIPS 2018
>
> [6] Mario Drumond, Louis Coulon, Arash Pourhabibi Zarandi, Ahmet Caner Yüzügüler, Babak Falsafi, Martin Jaggi: Equinox: Training (for Free) on a Custom Inference Accelerator. MICRO 2021
>
> [7] Sai Qian Zhang, Bradley McDanel, and HT Kung. Fast: Dnn training under variable precision block floating point with stochastic rounding. HPCA 2022.
>
> [8] Bita Darvish Rouhani, Daniel Lo, Ritchie Zhao, Ming Liu, Jeremy Fowers, Kalin Ovtcharov, Anna Vinogradsky, Sarah Massengill, Lita Yang, Ray Bittner, et al. Pushing the limits of narrow precision inferencing at cloud scale with Microsoft floating point. NeurIPS 2020.
>
>
>
> ---
>
>
> ## Q2: Orthogonality terminology
>
> We apologize for the confusion caused by the conventional interpretation of "orthogonality" in functional analysis. Thus, we propose renaming Definition 3.4 as “orthogonality/orthogonal in compression” to avoid misinterpretations. We revised the mentioned text in the paper. We are open to any other suggestions.

---

> ### Author Response · Authors · 2024-11-19
> **[Part 4/5] Clarifications on Theorems, Compression Methods, and Experimental Setup**
>
> ## Q5: Theorem 3.7 analyzes the error for the suboptimal order
>
> Theorem 3.7 is of interest because it provides a deeper understanding of the impact of the suboptimal order (quantization before sparsity) by quantifying the additional error. While Theorem 3.5 shows that applying sparsity before quantization (S$\rightarrow$Q) does not introduce any additional error, and Theorem 3.6 establishes that applying quantization before sparsity (Q$\rightarrow$S) may introduce additional error for at least one input $x$, Theorem 3.7 goes further by deriving an upper bound on this additional error for all possible inputs $x$ and demonstrates that this additional error is not negligible.
>
> ---
>
>
> ## Q6: Lower vs. Upper bound
>
> We apologize for any confusion caused by the terminology. We have updated the term to 'Orthogonality Threshold' to better reflect its purpose and provided an additional explanation (L310-314, Page 6). We are open to any other suggestions.
>
> ---
>
>
> ## Q7: Clarification of “lower value”
> We apologize for the confusion. By "lower values," we refer to the values of the model quality metric. For instance, metrics like *perplexity* are better when their values are lower ($\downarrow$ better). Conversely, for metrics like *accuracy*, higher values indicate better performance ($\uparrow$ better). We revised this sentence in the paper (L310-L315, Page 6) to clarify this point. Thank you for highlighting this.
>
> ---
>
>
> ## Q8: Sparse fine-tuning method
>
> We fine-tune our models, starting from the pre-trained dense checkpoints and applying weight sparsification by removing the weights with the smallest magnitudes at each training iteration. This approach is widely used for sparse fine-tuning. The learning schedule we follow is discussed in detail in Section 2.4.6, General Sparse Deep Learning Schedules, in the paper by Torsten Hoefler et al. [1]
>
> We included additional details regarding our finetuning setup and updated the citations in the revised manuscript (Section 4 Experimental Methodology and Results, Pages 6-7).
>
> [1] Torsten Hoefler, Dan Alistarh, Tal Ben-Nun, Nikoli Dryden, Alexandra Peste: Sparsity in Deep Learning: Pruning and growth for efficient inference and training in neural networks. Journal of Machine Learning Research 2021.
>
> ---
>
>
> ## Q9: Quantization method clarification
>
> We use max-scaled block-wise quantization, and the numerical formats we employ are mentioned and cited in the manuscript (Section 4 Experimental Methodology and Results - Page 7 - Lines 327-330).
>
>
> ---
>
> ## Q10: Clarification on fine-tuning a quantized model
>
> In our experiments, weights and activations are converted to their quantized representations during both the forward and backward passes of the fine-tuning process. However, we store master weights and perform weight updates in full precision without imposing restrictions on the resulting weight values. This approach is valid as we aim to adapt the model to compression during inference. We follow the prior work [1, 2].
>
> [1] Bita Darvish Rouhani et al.: Microscaling Data Formats for Deep Learning. arXiv 2023.
>
> [2] Mario Drumond, Tao Lin, Martin Jaggi, Babak Falsafi. Training DNNs with Hybrid Block Floating Point. NeurIPS 2018.
>
> ---
>
> ## Q11: Experimental set-up
>
> We summarize the Experimental setup in Table 6 in Appendix E (Page 20).
>
> ---
>
>
> ## Q12: Figure 1 in the “Stronger Generalization Bounds …” paper
>
> Figure 1 in our paper differs from Figure 1 in [1], representing the error across layers, where all layers are quantized and sparsified in a specific order. In contrast, Figure 1 in [1] illustrates the error when noise is injected exclusively into a single layer.
> However, Figure 2 in Appendix I (Page 23) is more comparable to what is depicted in [1]. Initially, Figure 2 showed the absolute error propagation rather than the relative error. We agree that the overall distribution of activations changes, and this must be considered when analyzing error dynamics. To address this, we updated the figure to illustrate the propagation of the relative error.
> In the revised version, Figure 2 shows that the relative error either increases, remains constant, or decreases slightly, which still differs from [1]. This discrepancy arises because the models studied in our paper, unlike VGG, incorporate skip connections that allow the error to propagate more freely throughout the network.
>
> [1]   Sanjeev Arora, Rong Ge, Behnam Neyshabur, Yi Zhang. Stronger Generalization Bounds for Deep Nets via a Compression Approach. ICML 2018.

---

> ### Author Response · Authors · 2024-11-19
> **[Part 5/5] Clarifications on Theorems, Compression Methods, and Experimental Setup**
>
> ## Q13:  Clarification of Effective Bitwidth
>
> We apologize for the confusion. By "effective bitwidth," we mean the practical precision of a quantization scheme, which depends on how bits are shared across elements. For instance, in block-wise quantization, where a scaling factor is shared across multiple elements, the average bitwidth per element is reduced compared to element-wise formats. Rouhani et al. refer to this concept as “average bits per element” [1]. To make the text more clear, we rephrase this sentence as the following: “Ideally, practitioners aim to maximize compression (increase sparsity ratio and/or reduce the average bitwidth per element). ”
>
> [1] Bita Darvish Rouhani et al.: With Shared Microexponents, A Little Shifting Goes a Long Way. ISCA 2023.
>
>
>
> ---
>
>
> ## Q14: Clarification of TOPS/mm2
>
> "TOPS" stands for Tera Operations Per Second, a measure of computational throughput. "mm²" refers to the silicon area of the hardware chip. Together, "TOPS per mm²" quantifies arithmetic density, representing the computational throughput relative to the physical area of the hardware. This metric is commonly used to evaluate the efficiency of AI accelerators and hardware platforms [1,2].
>
> [1] Rouhani et al. Pushing the Limits of Narrow Precision Inferencing at Cloud Scale with Microsoft Floating Point. NeurIPS 2020.
>
> [2] Mario Drumond, Tao Lin, Martin Jaggi, Babak Falsafi. Training DNNs with Hybrid Block Floating Point. NeurIPS 2018.

---

> ### Author Response · Authors · 2024-11-25
> **Follow-Up on Discussion**
>
> Dear Reviewer e8N1,
>
> Thank you once again for your detailed and constructive feedback. We appreciate the time and effort you’ve dedicated to reviewing our work and providing insightful questions. These questions quite helped us to further improve our paper.
>
> We wanted to follow up as the discussion period is ending soon. We have carefully addressed all the questions you raised in the initial review and provided a detailed response (five parts above). If there are any additional points of clarification, we would be happy to address them.
>
> We completely understand how demanding this period can be and deeply value your engagement. If there’s anything further we can clarify or improve, please don’t hesitate to let us know.
>
> Thank you for your time and consideration.
>
> Best regards,
>
> The Authors

---

> > ### Comment · Reviewer_e8N1 · 2024-11-26
> >
> > The responses to my concerns are reasonable, and I have adjusted the score to a 6. However, I did not rate it an 8 because I believe the theoretical analysis—central to the paper's primary contribution—has a limited scope, being restricted to magnitude pruning without fine-tuning.

---

> > > ### Author Response · Authors · 2024-12-02
> > > **Thank you!**
> > >
> > > Thank you for recognizing the contributions of our work and your recognition of our efforts to address your comments. We appreciate your adjustment to the score. We are grateful for your engagement with our work and your constructive perspective. Thank you for helping us refine our research and for highlighting valuable avenues to explore.

---

### Official Review · Reviewer_j3KK · 2024-11-01

**Soundness:** 2
**Presentation:** 3
**Contribution:** 3
**Rating:** 8
**Confidence:** 4

**Summary:**

This paper studies the interaction between weight sparsity, weight quantization, and activation quantization in small-to-moderate sized LLMs, ViTs, and CNNs. The authors prove and demonstrate empirically that these methods cannot be considered as purely orthogonal compression modalities under the orthogonality definitions proposed in the paper. Specifically, the authors show that the composition of these strategies is order-dependent and the combined error incurred generally exceeds the sum of the errors produced by applying each method independently.

**Strengths:**

* A timely and important topic as sparsity and quantization are promising compression strategies for the large model scales popular today.
* The paper includes a comprehensive summary of relevant literature.
* The proofs are relatively easy to follow and explained in an intuitive manner by the authors in the main text.
* Empirical results generally appear to support the theoretical findings.
* While many works have studied the combination of sparsity and quantization, this is the first that I am aware of to rigorously consider the interplay between these methods in detail.
* Empirical experiments include both LLMs and vision models.
* Extensive supplementary info includes an analysis of several leading SOTA methods from LLM pruning and quantization literature.

**Weaknesses:**

Overall I am leaning towards accept; however, some concerns regarding the empirical experimental design causes me to doubt the applicability of the results to more general settings:

* The primary metrics considered in the empirical results are perplexity or cross-entropy loss. While these are certainly reasonable proxies for downstream task performance, they are not perfectly correlated. While some accuracy metric for CV models was included in the appendices, it would be beneficial to extend this to downstream tasks for LLMs such as the OpenLLM v1 leaderboard evaluation tasks. It has been shown previously that PPL and CE can be particularly misleading metrics for quantized and sparse models [1].
* The experimental design for Section 4.1 is potentially concerning. If I understand the described process correctly, in the Q->S case the pretrained models are pruned and quantized before each forward pass (i.e., instantaneous masking and quantizing). Are the parameters themselves stored as dense fp32 tensors during this process and quantization is simulated similar to QAT approaches? Are the optimizer states left in fp32? The authors note issues with training dynamics in the Q->S case in Appendix A and my concern is that this could be related to numerical precision issues during fine-tuning rather than providing a reliable comparison on the order of compression. Adding a more detailed summary of the fine-tuning approaches in the appendix would potentially clear up any misunderstandings on this point.
* In the Q->S case quantized activations are used but in the S->Q case it appears the full precision activations are used. It's unclear to me if the dramatic difference in performance is caused by the quantized activations during fine-tuning rather than the specific order of compression for the weights.


[1] A. Jaiswal, Z. Gan, X. Du, B. Zhang, Z. Wang, and Y. Yang, “Compressing LLMs: The Truth is Rarely Pure and Never Simple,” Oct. 02, 2023, arXiv: arXiv:2310.01382. doi: 10.48550/arXiv.2310.01382.


### Suggestions / Typos:
* Defining “tensor and dot-product levels” earlier in the text would improve the reader's understanding. Specifically it may be worthwhile to relate these terms to “weights” and “activations” respectively.  I note that activations / dot-products are also represented as tensors.
* On L68, the authors refer to the challenge of quantizing LLMs due to outliers in “tensor distributions” and reference the smoothquant paper. This should be corrected to “dot-product outliers” as the challenge typically arises from outliers in the activations, not the weights (which instead follow a more gaussian-like distribution typically).
* I suggest separating references for fine-grained (N:M and similar) and structured (neuron-level or larger NN components) sparsity in the related work discussion on L115. In particular, it would be beneficial to introduce N:M sparsity before it appears in section 3.
* L469: state-of-the-arts -> state-of-the-art

**Questions:**

* In the Q->S case, the authors make the argument that this ordering may lead to additional errors when two otherwise unequal weights in the non-quantized precision are set to the same value once quantized. This is an intuitive conclusion but it would be interesting to ground this discussion in empirical evidence of the proportion of weights that this affects, on average, in a pre-trained model.
* Are the pretrained LLMs obtained from the base models or instruct-tuned variants? Making this explicit in the paper would be beneficial.
* L312 states that all linear layers were compressed for LLMs. Can you confirm that this included the lm-head, but not the encoder which is typically implemented as an embedding?
* Table 10 values for 1:4 are counter to typical intuition that higher sparsities generally perform worse. Could the authors confirm that this is 1:4 and not 3:4 sparsity?

---

> ### Author Response · Authors · 2024-11-19
> **[Part 1/2] Clarifications on Experiments, Fine-Tuning, and Compression Details**
>
> Dear Reviewer j3KK,
>
>
> We appreciate your comments and positive assessment of our work. We believe all your questions can be addressed within this discussion period. Please find our responses below, and feel free to ask further questions if anything remains unclear.
>
> ---
>
>
> ## W1: PPL and CV might be misleading. Add OpenLLM v1 experiments
>
> Thank you for raising this point. We agree that adding more downstream tasks for LLMs further strengthens our empirical results. To address this, we have included additional experiments in Appendix R (Page 33), presenting results for OPT-125M and Llama-3-8B across several OpenLLM v1 tasks under both compression orders (S$\rightarrow$Q and Q$\rightarrow$S). These results consistently demonstrate that S$\rightarrow$Q outperforms Q$\rightarrow$S, further validating our theoretical findings.
>
> ---
>
> ## W2: Detailed explanation of the fine-tuning setup
>
> We would like to clarify that our fine-tuning setup follows the standard approach in prior work [1,2] on training and fine-tuning with max-scaled block-wise number formats. During the Q$\rightarrow$S fine-tuning process, all weight updates are performed in FP32, ensuring no restrictions on weight changes that could adversely affect convergence. Specifically:
>
> **Quantization Emulation:** Since our primary focus is inference, we emulate quantization to enable or disable it during specific fine-tuning phases. Quantization is applied to matrix multiplications during the forward and backward passes to assess the impact of Q$\rightarrow$S order. However, the weight update phase is performed entirely in FP32 to maintain convergence. This approach aligns with standard practices [1,2].
>
> **Master Weights:** Master weights are stored in FP32 throughout fine-tuning to preserve numerical precision during updates. This strategy ensures that any numerical precision limitations arising from quantized updates do not affect convergence.
>
> We have added a more detailed explanation of the fine-tuning strategy in Appendix D (Page 20) and the experimental setup in Appendix E (Page 20). Thank you for your attention to this aspect.
>
> [1] Rouhani, Bita Darvish, et al. Microscaling data formats for deep learning. arXiv 2023.
>
> [2] Mario Drumond, Tao Lin, Martin Jaggi, and Babak Falsafi. 2018. Training DNNs with hybrid block floating point. NeurIPS 2018.
>
> ---
>
> ## W3: Quantizing the activations
>
> We apologize for the confusion and would like to clarify that, in both Q$\rightarrow$S and S$\rightarrow$Q cases, weights and activation tensors are quantized before matrix multiplication operations (see Line 350 - Page 7). This strategy ensures a consistent comparison between the two compression orders, as the same quantization approach is applied uniformly to both activations and weights in both scenarios.
>
> ---
>
> ## Q1: Empirical evidence of the proportion of weights that $Q \rightarrow S$ reordering affects
>
> We agree that our conclusion can be further strengthened with additional empirical evidence. To address this, we have added Appendix Q (Page 32), where we experimentally analyze the impact of quantization on the number of duplicates in tensors. Our findings indicate that quantization significantly increases the number of duplicates in a tensor. The number of unique elements in a block drops by up to 64% after quantization, which increases the chance of pruning a wrong element in the case of applying $Q \rightarrow S$ (Reordering effect description in Lines 1338-1340, Page 25).
>
> ---
>
>
> ## Q2:  Base models or instruct-tuned variants?
>
> The pretrained LLMs used in our experiments are base (general-purpose) models, not instruct-tuned variants. We made this explicit in the revised manuscript to ensure clarity for readers and reproducibility of the results (Lines 322-323, Page 6). If you find it necessary, we would be happy to extend the results to include instruction-tuned models in the final version of the paper.

---

> ### Author Response · Authors · 2024-11-19
> **[Part 2/2] Clarifications on Experiments, Fine-Tuning, and Compression Details**
>
> ## Q3: Which layers are compressed?
>
> We confirm that we do not compress the *lm-head* or *embedding layers* in our LLM experiments.
> We updated the manuscript to explicitly state that the *lm-head* and *embedding layers* are excluded from compression (Line 347, Page 7).
>
> Practitioners typically exclude these layers from compression due to their sensitivity to compression and significant impact on model quality [1, 2, 3]. Moreover, the memory savings from the compression of these layers are generally negligible compared to the potential performance degradation. For instance, in Llama-3.1-70B, the combined parameters in the lm-head and embed_tokens layers total approximately 2 billion, representing only 2.9% of the total model weights. In smaller models, such as Llama-3.1-8B, these layers account for about 1 billion parameters or 13% of the total weights. Compressing these layers would not yield substantial compression rates but could compromise performance.
>
> [1] Lee, Changhun, et al. "OWQ: Outlier-aware weight quantization for efficient fine-tuning and inference of large language models." AAAI 2024.
>
> [2] Frantar, Elias, et al. "OPTQ: Accurate quantization for generative pre-trained transformers." ICLR 2022.
>
> [3] Frantar, Elias, and Dan Alistarh. "Sparsegpt: Massive language models can be accurately pruned in one shot." ICML 2023.
>
> ---
>
>
> ## Q4: 1:4 or 3:4?
>
> We apologize for the confusion and thank you for bringing this to our attention. In the revised paper, Table 10 became Table 11. Previously, 1:4 sparsity was used to refer to 1 nullified element out of 4, but to ensure consistency, we have updated it to refer to 3:4 sparsity.
>
> We acknowledge that there is no consensus in the literature regarding whether N:M sparsity refers to N nullified elements or N non-nullified elements [1,2]. To clarify, we have explicitly stated our notation in the revised manuscript (Lines 330-331, Page 7) and ensured consistent usage across all tables (Tables 8, 11 and 12).
>
> [1] Pool, Jeff, and Chong Yu. "Channel permutations for n:m sparsity." NeurIPS 2021.
>
> [2] Zhou, Aojun, et al. "Learning n:m fine-grained structured sparse neural networks from scratch." ICLR 2021.
>
>
> ---
>
> ## Suggestions / Typos:
>
> Thank you for finding the typos. We fixed all of them. Below, please find our responses to your other suggestions:
>
> **(1) Tensor and Dot-Product Levels** We have modified the text at the beginning of Section 3 to make our notation clear. Please refer to lines 153-156 in the revised paper.
>
>
> **(2) Dot-Product Outliers** We revised the text on Line 68 to explicitly mention "dot-product outliers in activation tensors" while retaining the broader context of tensor distributions.
>
> **(3) N:M Sparsity Description** We have incorporated the related work previously in the Appendix into the main text, providing a clearer introduction to N:M sparsity before it is discussed in Section 3 (Also, in line with Reviewer KXV4’s feedback). This revision ensures that readers are familiar with the distinctions and context before encountering these concepts in later sections.

---

> ### Author Response · Authors · 2024-11-25
> **Follow-Up on Discussion**
>
> Dear Reviewer j3KK,
>
> Thank you once again for your detailed and constructive feedback. We appreciate the time and effort you’ve dedicated to reviewing our work and providing insightful questions. These questions quite helped us to further improve our paper.
>
> We wanted to follow up as the discussion period is ending soon. We have carefully addressed all the questions you raised in the initial review and provided a detailed response (two parts above). If there are any additional points of clarification, we would be happy to address them.
>
> We completely understand how demanding this period can be and deeply value your engagement. If there’s anything further we can clarify or improve, please don’t hesitate to let us know.
>
> Thank you for your time and consideration.
>
> Best regards,
>
> The Authors

---

> > ### Comment · Reviewer_j3KK · 2024-11-25
> >
> > I thank the authors for their detailed rebuttal, clarifications, and updated manuscript. My original concerns have been adequately addressed and as such I have raised my score to 8.

---

> > > ### Author Response · Authors · 2024-12-02
> > > **Thank you!**
> > >
> > > Thank you for engaging with our work and for your thoughtful feedback throughout the review process. We greatly appreciate your updated score and your recognition of our efforts to address your comments.

---

### Official Review · Reviewer_55F2 · 2024-11-04

**Soundness:** 4
**Presentation:** 4
**Contribution:** 3
**Rating:** 8
**Confidence:** 4

**Summary:**

This paper provides a comprehensive theoretical and empirical investigation into the interplay between sparsity and quantization, two widely used model compression techniques. The authors mathematically prove that sparsity and quantization are non-orthogonal operations, meaning their combined use introduces compounded errors beyond those incurred by each method independently. They further derive the optimal order of applying sparsity before quantization (S→Q) to minimize additional errors. These theoretical findings are validated through extensive experiments on large language models (OPT, LLaMA), vision transformers (ViT), and convolutional neural networks (ResNet). The paper also introduces the novel "orthogonality bound" metric to efficiently estimate the performance of sparse-quantized models without expensive retraining.

**Strengths:**

- The paper makes significant theoretical contributions by proving the non-orthogonality of sparsity and quantization and deriving the optimal $S\to Q$ order. These insights challenge conventional assumptions and provide valuable guidance for model compression.
- The mathematical analysis is rigorous and comprehensive, covering tensor-level and dot product-level errors.
- The experimental results are extensive, spanning diverse models (OPT, LLama, ResNet, ViT) and settings.
- The orthogonality bound metric seems like a useful tool for practitioners.
- Overall the paper is well-structured, with clear definitions, detailed appendices, and informative tables.

**Weaknesses:**

- While the experiments cover a range of models and settings, the datasets used (WikiText2, ImageNet1k) are relatively small and few. Evaluating on larger, more challenging datasets would further strengthen the findings.
- The paper does not explore the impact of different sparsity patterns (e.g., block-wise sparsity) or more advanced quantization schemes.

**Questions:**

- Evaluating the findings on larger, more diverse dataset would be nice.
- It would be interesting to see how the optimal $S\to Q$ order and orthogonality bound extend to other sparsity patterns and quantization schemes. Can the authors comment on the generality of their findings in this regard?

---

> ### Author Response · Authors · 2024-11-19
> **Dataset Expansion and Compression Methodology**
>
> Dear Reviewer 55F2,
>
> We appreciate your comments and positive assessment of our work. We believe all your questions can be addressed within this discussion period. Please find our responses below, and feel free to ask further questions if anything remains unclear.
>
>
> ---
> ## W1 - Q1: Evaluation of larger and more diverse datasets
>
> Thank you for your suggestion. We agree that incorporating more datasets further strengthens our empirical results. To address this, we have added additional experiments in Appendix R (Page 33), reporting results for OPT-125M and Llama-3-8B across several OpenLLM v1 tasks under both compression orders (S$\rightarrow$Q and Q$\rightarrow$S). These results demonstrate that S$\rightarrow$Q consistently outperforms Q$\rightarrow$S, further validating our theoretical findings.
>
> ---
>
> ## W2 - Q2: Impact of other sparsity patterns and quantization schemes
>
> Thank you for your question. Our study focuses on magnitude-based sparsity and block-wise quantization, as these are widely adopted in the field. We agree that further exploring different sparsity patterns and other quantization techniques is an important avenue for future work. We will highlight this direction in the discussion section of the revised manuscript. Please see the detailed responses below.
>
> **Generality of Theoretical Results:** Our theoretical analysis examines the interplay between sparsity and quantization, highlighting compounded errors caused by their non-orthogonality. These insights are not inherently tied to specific sparsity patterns (e.g., block-wise sparsity) or quantization schemes. Instead, they provide a framework that can be extended for other compression methods in future work.
>
> **Practical Significance of Block-Wise Quantization:**
> Max-scaled block-wise quantization is commonly used in research and industry due to its ability to reduce memory footprint, improve arithmetic density (especially in sub-8-bit regimes), and maintain accuracy with minimal hardware overhead. This characteristic makes it ideal for modern ML accelerators, as demonstrated by prior work (e.g., Drumond et al. [5,6], Zhang et al. [7], and Rouhani et al. [3,4,8]). Additionally, vendors under the Open Compute Project (OCP) [2,3,4] have proposed block-wise scaled numerical formats like MXFP, which use shared microexponents for fine-grained scaling. These techniques have been adopted in modern hardware platforms, such as the upcoming Nvidia Blackwell GPUs [1]. We have updated the Related Work section to highlight the prevalence and practical importance of these compression methods.
>
> **Empirical Validation:** While our study does not explicitly evaluate other sparsity patterns or quantization schemes, we empirically evaluated the interplay of sparsity and quantization using other methods. For instance, Table 7 (Page 21) in the appendix includes results for SparseGPT and WANDA sparsity methods combined with block-wise quantization (more details in Appendix F - Page 20).
> In addition, Tables 10 and 11 (Page 31) present results for combining different sparsity methods and GPTQ (more details in Appendix N - Page 31). These additional empirical results suggest that the interplay between sparsity and quantization is a fundamental consideration, regardless of the specific methods used.
>
> ---
>
> [1] https://www.nvidia.com/en-us/data-center/tensor-cores/#blackwell
>
> [2]https://www.opencompute.org/blog/amd-arm-intel-meta-microsoft-nvidia-and-qualcomm-standardize-next-generation-narrow-precision-data-formats-for-ai
>
> [3] Bita Darvish Rouhani et al.: Microscaling Data Formats for Deep Learning. arXiv 2023.
>
> [4] Bita Darvish Rouhani et al.: With Shared Microexponents, A Little Shifting Goes a Long Way. ISCA 2023.
>
> [5] Mario Drumond, Tao Lin, Martin Jaggi, Babak Falsafi: Training DNNs with Hybrid Block Floating Point. NeurIPS 2018.
>
> [6] Mario Drumond, Louis Coulon, Arash Pourhabibi Zarandi, Ahmet Caner Yüzügüler, Babak Falsafi, Martin Jaggi: Equinox: Training (for Free) on a Custom Inference Accelerator. MICRO 2021.
>
> [7] Sai Qian Zhang, Bradley McDanel, and HT Kung. Fast: DNN Training Under Variable Precision Block Floating Point with Stochastic Rounding. HPCA 2022.
>
> [8] Bita Darvish Rouhani, Daniel Lo, Ritchie Zhao, Ming Liu, Jeremy Fowers, Kalin Ovtcharov, Anna Vinogradsky, Sarah Massengill, Lita Yang, Ray Bittner, et al. Pushing the Limits of Narrow Precision Inferencing at Cloud Scale with Microsoft Floating Point. NeurIPS 2020.

---

> > ### Comment · Reviewer_55F2 · 2024-11-23
> > **Official Comment by Reviewer 55F2**
> >
> > Thank you for your reply. All my concerns have been solved, and I would like to maintain by score of 8.

---

> > > ### Author Response · Authors · 2024-11-25
> > > **Thank you!**
> > >
> > > Thank you once again for your review and for letting us know that all of your concerns have been resolved. We truly appreciate your positive evaluation and the time you’ve dedicated to our work!

---

### Official Review · Reviewer_KXV4 · 2024-11-04

**Soundness:** 3
**Presentation:** 4
**Contribution:** 3
**Rating:** 8
**Confidence:** 4

**Summary:**

The paper explores the relationship between two widely used compression techniques: sparsity and quantization. Specifically, it demonstrates that these techniques are not independent of one another; the order in which they are applied can significantly impact the results. Additionally, their combination can lead to error propagation, with accumulated errors affecting consecutive layers. The study draws from both theoretical analysis and experimental results conducted on large, modern neural networks.

**Strengths:**

- The paper covers an interesting and timely topic. Given the increasing size of parameters in pre-trained models, there is growing interest in techniques such as quantization and sparsity. Providing both analytical and empirical insights into the relationship between these techniques is valuable, especially as they are often studied separately.The findings in this paper, such as the optimal order for applying sparsity and quantization and the established upper bounds, can offer practical guidance for researchers in this area.
- The paper effectively demonstrates the non-orthogonality of sparsity and quantization, determining the optimal sequence for applying these transformations through theoretical analysis, supported by empirical studies on large, modern networks.
- The work is well-written, easy to follow, and enjoyable to read.

**Weaknesses:**

- In the experiments section, the results appear promising and generally align with the theoretical findings. However, it is unclear whether the reported results represent averages of multiple runs or single-run outcomes. If they are averages, what are the standard deviations?
- Additionally, I believe the related work section should remain in the main body of the paper, particularly since there is available space before reaching the 10-page limit. Moving it to the appendix could diminish its visibility and importance.

**Questions:**

- In Table 2, what do the bold-out results represent? This should be explained in the caption.
- In Table 2: perhaps it would be beneficial to show the delta to the sparsity 0% instead/additionally  (e.g. in the appendix)?

Overall, I find the topic of this paper both interesting and potentially valuable to researchers focusing on sparsity and quantization. The claims and theorems are clearly articulated (I briefly reviewed the details of Theorems 3.5, 3.6, 3.7, and 3.9 in the appendix), and the empirical evaluation, while primarily centered on magnitude-based sparsity, is compelling and conducted across various models and tasks. I believe the strengths of the paper outweigh weaknesses (in fact, I do not have significant concerns regarding weaknesses). Therefore, I am inclined to recommend acceptance.

---

> ### Author Response · Authors · 2024-11-19
> **Clarifications on Results, Related Work, and Table 2 Updates**
>
> Dear Reviewer KXV4,
>
> We appreciate your comments and positive assessment of our work. We believe all your questions can be addressed within this discussion period. Please find our responses below, and feel free to ask further questions if anything remains unclear.
>
>
> ---
>
>
> ## W1: Are the results averages of multiple runs or single-run?
>
> The reported results in the experiments section are single-run outcomes, as we were constrained by limited compute resources. However, to assess robustness, we conducted additional analysis on the OPT-125M model using the HBFP8/6 number format with three distinct random seeds. These results, including standard deviations, are presented in Appendix H (Page 22) and summarized in Table 7 (Page 21).
>
> ---
>
> ## W2: Related work should remain in the main text
> We appreciate your feedback to improve the quality of our work. We agree with your suggestion and have updated the main body accordingly.
>
> ---
>
> ## Q1: Explain the bold part in Table 2
>
> We apologize for the confusion. The bold values in Table 2 represent the lower perplexity (lower is better) between the measured loss and the orthogonality bound for a particular compression configuration on a specific model (e.g., 2:4 sparsity with HBFP8 for OPT-125M). We updated the table caption to clarify this explicitly.
>
> ---
>
> ## Q2: Delta values in Table 2
>
> We have updated the paper to include the delta relative to the 0% sparsity results in Table 2.

---

> > ### Comment · Reviewer_KXV4 · 2024-11-23
> >
> > I thank the authors for their reply.
> >
> > Regarding W1: I understand. While I would have preferred results from multiple runs, the consistency of observations across various datasets and quantization setups convinces me that the technical evaluation is sound.
> >
> > Regarding W2, Q1, Q3: Thank you for addressing this.
> >
> > Overall, I am satisfied with the authors' response and will maintain my score, as it already appropriately reflects my positive assessment of the paper.

---

> > > ### Author Response · Authors · 2024-11-23
> > > **Additional multirun results**
> > >
> > > Dear Reviewer KXV4,
> > >
> > > Thank you for your response and for acknowledging the soundness of our evaluation.
> > >
> > > We acknowledge your concern and are happy to include additional multirun results in the final version, given their computational demand and the availability of our resources, to further reinforce the consistency of our findings.
> > >
> > > Thank you again for your time and constructive comments, which have helped improve our work.
> > >
> > > Best regards,
> > >
> > > The Authors

---

### Author Response · Authors · 2024-11-20
**General Response: Addressing Feedback and Highlighting Revisions**

Dear Area Chair and Reviewers,

Thank you for taking the time to review our manuscript and providing thoughtful and constructive feedback. We have submitted a revised version of our paper, incorporating all of the feedback, with new changes highlighted in $\textcolor{blue}{blue}$.

We greatly appreciate the positive remarks from the reviewers, highlighting the following strengths of our work:

1. Addressing an interesting and timely topic with significant potential value for researchers focused on sparsity and quantization [[Reviewer KXV4](https://openreview.net/forum?id=wJv4AIt4sK&noteId=lDdVaHFtOv)],

2. Delivering significant theoretical contributions through rigorous mathematical analysis and comprehensive experimental results [[Reviewer 55F2](https://openreview.net/forum?id=wJv4AIt4sK&noteId=SthsrutlKD)],

3. Being the first work to rigorously examine the interplay between sparsity and quantization in detail, addressing a critical and timely topic [[Reviewer j3KK](https://openreview.net/forum?id=wJv4AIt4sK&noteId=bjNgWZDVmI)].

4. Investigating an overlooked problem in the sparsity and quantization literature with theoretical proofs, derivations, and rigorous empirical studies that substantiate our claims [[Reviewer e8N1](https://openreview.net/forum?id=wJv4AIt4sK&noteId=ozGKzq1QaT)].

Your insightful feedback has been instrumental in improving our manuscript. We believe we have addressed all the questions and comments. In response, we have made the following  updates and improvement to the manuscript:

- **Additional Empirical Results** $\rightarrow$ Added additional evaluation results on OpenLLM (four more tasks) across two models (OPT-125M, Llama-3-8B) — see Tables 15-16 Pages 36-37.

- **Finetuning Dynamics** $\rightarrow$ Provided further analysis to align finetuning dynamics with our theoretical framework, including zero-shot results (Tables 15-16 Pages 36-37) and weight distribution analysis (Appendix S Pages 34-35).

- **Clarifications and Revisions** $\rightarrow$ Clarified all points regarding our analysis and experimental details.

We appreciate your valuable feedback and the opportunity to further improve our work. Please let us know if there are any remaining questions or additional suggestions. We are eager for a constructive and insightful discussion with the reviewers during this period and welcome any further input you may have.

Best regards,

The Authors

---

### Meta-Review · Area_Chair_ACvn · 2024-12-19

**Metareview:**

In this paper the authors consider the error introduced by various model compression schemes for deep networks, such as sparsifying or quantizing the numerical precision of the weights.  While both approaches are commonly used, here the authors consider the combination of the two methods.  In particular, the authors provide theoretical results showing that the ordering of the sparsification and quantification operators matters, with more error occurring if quantification is done before sparsification, and that even if applied in correct order the errors from applying the two techniques in combination can be greater than the sum of the errors from applying each technique independently.

While the basic message of these results are perhaps intuitive and to be expected, the reviewers are largely in agreement that having a theoretical characterization of these effects is a worthwhile contribution.  Some reviewers still have concerns that the applicability of the work may be limited as the results only apply to magnitude based sparsity pruning without fine-tuning, whereas methods like SparseGPT and WANDA may have stronger performance in settings where fine-tuning is too costly.  While this is potentially a valid concern, this seems to be a reasonable avenue for future work, and I agree with the overall consensus of the reviewers to accept the paper for publication.

**Additional Comments On Reviewer Discussion:**

The authors were responsive to the reviewers' questions and concerns during the rebuttal period, with many of the reviewers raising their scores or maintaining already positive scores.

---

### Decision · Program_Chairs · 2025-01-22

Accept (Spotlight)